# Transcriptional profiling of human microglia reveals grey–white matter heterogeneity and multiple sclerosis-associated changes

Marlijn van der Poel[1], Thomas Ulas[2], Mark R. Mizee[1], Cheng-Chih Hsiao[3], Suzanne S. M. Miedema[1], Adelia[4], Karianne G. Schuurman[1], Boy Helder[3,5], Sander W. Tas[3,5], Joachim L. Schultze[2,6], Jörg Hamann[1,3] & Inge Huitinga[1]

Here we report the transcriptional profile of human microglia, isolated from normal-appearing grey matter (GM) and white matter (WM) of multiple sclerosis (MS) and non-neurological control donors, to find possible early changes related to MS pathology. Microglia show a clear region-specific profile, indicated by higher expression of type-I interferon genes in GM and higher expression of NF-κB pathway genes in WM. Transcriptional changes in MS microglia also differ between GM and WM. MS WM microglia show increased lipid metabolism gene expression, which relates to MS pathology since active MS lesion-derived microglial nuclei show similar altered gene expression. Microglia from MS GM show increased expression of genes associated with glycolysis and iron homeostasis, possibly reflecting microglia reacting to iron depositions. Except for *ADGRG1*/GPR56, expression of homeostatic genes, such as *P2RY12* and *TMEM119*, is unaltered in normal-appearing MS tissue, demonstrating overall preservation of microglia homeostatic functions in the initiation phase of MS.

[1] Neuroimmunology Research Group, Netherlands Institute for Neuroscience, Meibergdreef 47, 1105BA Amsterdam, The Netherlands. [2] Genomics and Immunoregulation, LIMES Institute, University of Bonn, Carl-Troll-Straße 31, 53115 Bonn, Germany. [3] Department of Experimental Immunology, Amsterdam University Medical Centers, University of Amsterdam, Meibergdreef 9, 1105AZ Amsterdam, The Netherlands. [4] Netherlands Brain Bank, Netherlands Institute for Neuroscience, Meibergdreef 47, 1105BA Amsterdam, The Netherlands. [5] Department of Clinical Immunology and Rheumatology, Amsterdam Rheumatology and Immunology Center, Amsterdam University Medical Centers, University of Amsterdam, Meibergdreef 9, 1105AZ Amsterdam, The Netherlands. [6] PRECISE Platform for Single Cell Genomics and Epigenomics, German Center for Neurodegenerative Diseases, University of Bonn, Sigmund-Freud-Street 27, 53127 Bonn, Germany. These authors contributed equally: Marlijn van der Poel, Thomas Ulas. These authors jointly supervised this work: Jörg Hamann, Inge Huitinga. Correspondence and requests for materials should be addressed to J.H. (email: j.hamann@amc.uva.nl) or to I.H. (email: i.huitinga@nin.knaw.nl)

Multiple sclerosis (MS) is a chronic neuroinflammatory disease, characterized by demyelination and neuroaxonal damage, that leads to the formation of lesions throughout the central nervous system (CNS)[1,2]. Grey matter (GM) and white matter (WM) lesions show pathological differences, demonstrated by little infiltration of lymphocytes and a low number of activated microglia in GM cortical lesions when compared to WM lesions[3,4]. Microglial immune activation is present in MS lesions, and microglia play a role as phagocytes in demyelination, yet this role remains far from understood[5,6].

Microglia are brain-resident phagocytic cells that originate from the yolk sac during embryonic development[7] and are maintained through a constant turnover without influx of circulating monocytes during adulthood[8]. Microglia possess a unique transcriptomic profile, compared to other cells of the CNS or tissue macrophages[9–11]. They play an essential role in maintaining brain homeostasis by surveying the brain and quickly responding to changes in their environment[12]. Environmental changes, like chronic neuroinflammation or neuronal damage, induce microglial state changes, resulting in altered functions to control damage and induce repair[13,14]. Recent studies describe a downregulation of homeostatic gene expression in relation to neuropathological conditions in animal models[15,16] and to aging in humans[17]. Microglia are further implicated as players in MS neuropathology by the finding that 48 out of 81 genes with single-nucleotide polymorphisms associated with MS are highly expressed by microglia[11]. Evidence for local environmental changes that lead to an altered, more alerted microglial phenotype has been found in perilesional areas in MS, where axons decorated with complement components attract clusters of microglia[18]. In addition, we previously reported an alerted phenotype of microglia isolated from post-mortem normal-appearing WM (NAWM) tissue from MS donors compared to control WM, as demonstrated by increased expression of CD45[19].

The role of microglia in lesion initiation in both GM and WM regions in MS is not yet elucidated. Therefore, we focused on normal-appearing MS tissue to find early alterations in microglia linked to putative lesion development. These regions are devoid of observable MS pathology, like loss of myelin proteins, but might harbor microglial state changes related to lesion formation. Several gene expression studies on NAWM MS tissue found evidence for alterations in expression of immunosuppressive as well as pro-inflammatory genes[20], and we reported upregulation of scavenger receptor and lipid metabolism genes in WM adjacent to chronic active MS lesions[21,22]. In addition, early signs of alterations in axonal integrity in normal-appearing tissue were shown by magnetic resonance imaging (MRI)[23]. Moreover, changes in myelin lipid composition in normal-appearing tissue have been described[24]. Interestingly, we previously found that myelin isolated from NAWM from MS brain donors was more efficiently phagocytosed compared to myelin from control donors by primary microglia in vitro[25].

We postulate that early pathological changes in normal-appearing MS tissue might be reflected by molecular changes in microglia, thereby elucidating mechanisms underlying lesion initiation. Therefore, we analyzed the transcriptional profile of human microglia isolated post-mortem from normal-appearing GM (NAGM) and NAWM brain regions of 11 control and 10 MS brain donors and identified their transcriptional profile by RNA sequencing. We here provide evidence for MS-related changes in microglia from normal-appearing tissue and profound intrinsic differences in the microglial transcriptome between GM and WM brain regions, identified by differential gene expression and pathway enrichment analyses.

## Results

**A common core signature for GM and WM microglia.** Human microglia were isolated from occipital cortex (GM) and corpus callosum (WM) post-mortem tissue of both control and MS donors using a validated protocol of subsequent enzymatic dissociation, density gradient separation, and magnetic bead sorting[26]. Details on clinical information and post-mortem variables of all donors are provided in Supplementary Data 1. Three MS donors were diagnosed with primary-progressive MS and 7 donors had secondary-progressive MS. The MS donors had a disease duration of on average 28.7 years, and based on histopathological analysis, 37% of total lesions in post-mortem brain are active (defined as active lesion load)[27]. MS groups did not differ from the control groups with respect to age, gender, peripheral inflammation, pH of cerebrospinal fluid (CSF), and RNA integrity number (RIN) but, due to post-mortem MRI during autopsy, had a post-mortem delay (PMD) of 3 h longer as compared to control donors (Table 1). However, we did not detect any significant correlation between PMD and MS-related differentially expressed (DE) genes, demonstrating that PMD had no effect on gene expression (Supplementary Figure 2). Immunohistochemistry performed on tissue used for microglia isolation showed ramified microglia morphology based on HLA-DR staining in control and MS donors. No differences in microglia activation between control and MS donors were found based on HLA-DR and CD68 staining, and no signs of demyelination were indicated based on proteolipid protein (PLP) staining (Supplementary Figure 3).

Low detection of transcripts specifically expressed in endothelial cells (ITM2A, ADGRL4, CLDN5), astrocytes (S100B, SOX9, GFAP), neurons (ENO2, MAP2), and oligodendrocytes (CSPG4, MOG, MAG) and high expression of transcripts commonly associated with microglia (CSF1R, CX3CR1, P2RY12) confirmed the purity of the obtained microglia population (Fig. 1a).

**Table 1 Donor characteristics for MS and control donors used for RNA sequencing**

| Diagnosis | Gender (M/F) | Age, years | PMD | pH CSF | PI | Active lesion load | Disease duration | MS type | Region | RIN |
|---|---|---|---|---|---|---|---|---|---|---|
| CON | 4/7 | 76.5 ± 4.1 | 6:06 ± 0:18 | 6.5 ± 0.08 | Yes: 6 No: 5 | | | | CC OC | 7.3 ± 0.4 7.0 ± 0.5 |
| MS | 6/4 | 66.7 ± 4.5 | 9:17 ± 0:18 | 6.4 ± 0.06 | Yes: 4 No: 6 | 0.37 ± 0.08 | 28.7 ± 3.53 | PP: 3 SP: 7 | CC OC | 8.1 ± 0.3 6.3 ± 0.8 |
| | n.s. | n.s. | p = 0.0001 | n.s. | n.s. | | | | | n.s. |

Data are represented as mean with SEM. Statistical testing: Gender and peripheral inflammation = Fisher's exact test; age, PMD, and pH CSF = unpaired t test; RIN = Kruskal–Wallis test
*Active lesion load* total number of active lesions/total number of all lesion types[27], *age* age at death (years), *CC* corpus callosum, *CON* non-neurological control, *CSF* cerebrospinal fluid, *disease duration* time between MS diagnosis until dead (years), *F* female, *M* male, *MS* multiple sclerosis, *OC* occipital cortex, *PI* peripheral inflammation, *PMD* post-mortem delay (h:min), *PP* primary progressive, *RIN* RNA integrity number, *SP* secondary progressive, *n.s.* not significant

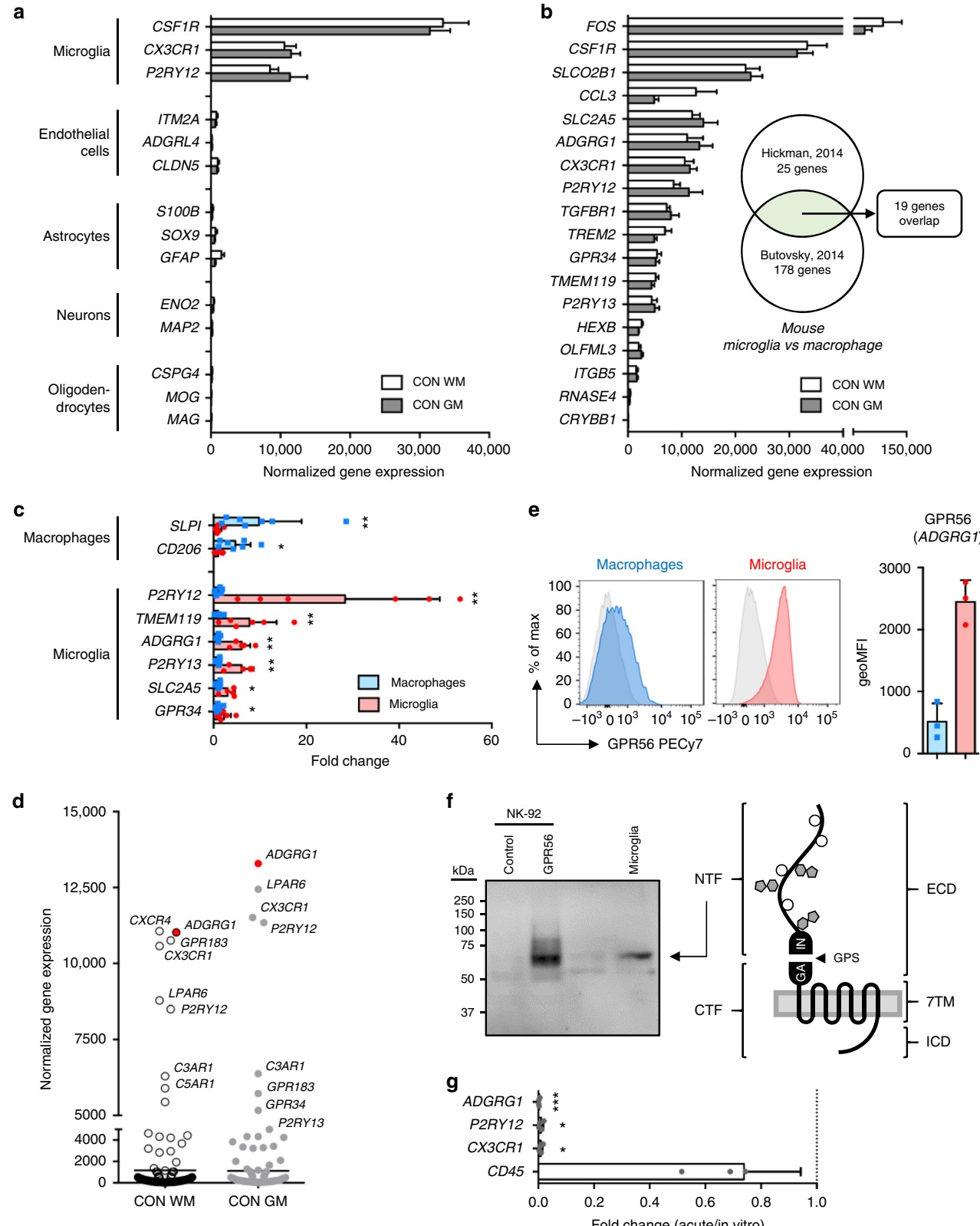

Several studies have reported a common signature of genes expressed by microglia, different from tissue-resident macrophages, in mice[28,29] and more recently also in humans[10,11]. Comparing our data set, we found 17 out of 19 top mouse microglia signature genes to be highly expressed in GM and WM human microglia (Fig. 1b)[28,29]. Furthermore, genes expressed by GM and WM microglia overlap with the top 30 genes found in two human microglia gene expression studies (Supplementary Figure 4)[10,11]. By comparing transcription levels in WM microglia with choroid plexus macrophages, we found a significant enrichment for *P2RY12* (fold change (FC) 28.36, $p = 0.001$), *TMEM119* (FC 7.71, $p = 0.005$), *ADGRG1* (FC 6.03, $p = 0.001$), *P2RY13* (FC 6.01, $p = 0.001$), *SLC2A5* (FC 3.03, $p = 0.01$), and *GPR34* (FC 2.26, $p = 0.04$) in microglia by quantitative reverse transcription polymerase chain

**Fig. 1** A common core signature for grey and white matter (GM and WM, respectively) microglia. **a** Expression of genes associated with microglia, endothelial cells, astrocytes, neurons, and oligodendrocytes obtained by RNA sequencing of GM ($n = 5$) and WM ($n = 11$) microglia. **b** Expression of 19 established mouse microglia signature genes that overlap between two datasets[28,29] is highly conserved in human microglia from both GM ($n = 5$) and WM ($n = 11$) tissue. **c** Expression of microglia and macrophage signature genes in WM microglia ($n = 6$) and choroid plexus macrophages ($n = 7$) determined by quantitative reverse transcription polymerase chain reaction (RT-qPCR). Mann–Whitney U test *$p < 0.05$, **$p < 0.01$. **d** Transcriptional expression of G protein-coupled receptors (GPCRs) in microglia reveals *ADGRG1* as top 3 abundant GPCR transcript in both GM ($n = 5$) and WM ($n = 11$) microglia. **e** Detection of GPR56 protein in WM microglia, but not in choroid plexus macrophages, by flow cytometry ($n = 3$). **f** Western blot analysis of GPR56 in NK-92 cells and WM microglia detects a 60-kDa band, corresponding with the presumed size of the N-terminal fragment (NTF). **g** In culture, microglia downregulate the expression of *ADGRG1* and other signature genes (*P2RY12*, *CX3CR1*) determined here by RT-qPCR at 4 days in vitro ($n = 4$). Unpaired t test: *$p < 0.05$, ***$p < 0.001$. Bars show mean with standard deviation

reaction (RT-qPCR). In contrast, the well-known macrophage genes *SLPI* (FC 9.80, $p = 0.002$) and *CD206* (FC 4.69, $p = 0.02$) were enriched in choroid plexus macrophages, compared to microglia (Fig. 1c).

The sensome of microglia comprises numerous G protein-coupled receptors (GPCRs), including the purinergic receptors *P2RY12* and *P2RY13*, the fractalkine receptor *CX3CR1*, and the orphan receptor *GPR34*[9]. We noticed that *ADGRG1*, encoding GPR56[30], is another top abundant GPCR transcript in human GM and WM microglia (Fig. 1d). In line with the known absence of *ADGRG1* gene expression in myeloid cells other than microglia[31,32], we detected GPR56 protein expression on acutely isolated primary WM microglia ($2449 \pm 348$), but hardly on choroid plexus macrophages ($514 \pm 295$) (Fig. 1e and Supplementary Figure 5). Western blot analysis of GPR56 in microglia revealed a 60-kD band (Fig. 1f), corresponding in size with the extracellular fragment of the processed receptor[33]. Corroborating the massive changes in microglia signature gene expression seen in microglia removed from their natural microenvironment[11,26], transcription of *ADGRG1* was completely lost after 4 days in culture (FC 0.15, $p = 0.0006$; Fig. 1g).

To conclude, the microglia we isolated from both GM and WM regions of control and normal-appearing MS post-mortem tissue express genes characteristic for microglia, including several GPCRs.

**Regional and MS-specific changes in gene expression.** A principal component analysis (PCA) showed clustering of four groups clearly distinguishing microglia from GM and WM brain regions (Fig. 2a). Less distinction was observed between microglia isolated from control and normal-appearing MS tissue. DE genes were defined among GM and WM microglia for both MS and control donors (Fig. 2b and Supplementary Data 2). The highest number of DE genes was observed between GM and WM regions, 453 genes in control, and 124 genes in microglia from MS donors. Remarkably, the number of DE genes in WM microglia compared to GM in control donors is higher compared to MS donors. When comparing the MS and control groups, a higher number of genes were differentially expressed in GM (93 genes) as compared to WM (38 genes) microglia. When comparing MS DE genes between GM and WM, hardly any overlap was observed; only two genes (*MRC1* and *LOC101927481*) were jointly regulated, demonstrating that MS-associated changes are region specific (Fig. 2c).

DE genes were displayed in volcano plots for all four comparisons (Fig. 2d–g). Top DE genes with the highest FC, highly expressed in GM microglia, were *TNFRSF25* and *CCL2*, which play a role in the "cytokine-mediated signaling" (GO-term analysis; in Supplementary Data 2). WM microglia more abundantly expressed genes involved in "chemotaxis" and "inflammatory response" (*CXCR4*, *ACKR1*, *GPNMB*, *NUPR1*). DE genes highly expressed in GM compared to WM microglia in the MS group (*TNFRSF25*, *CCL2*, *MRC1*) were mainly the

same as in the control group. In addition, inflammatory calgranulin genes *S100A12* and *S1000A9* were significantly higher expressed in GM compared to the WM MS microglia. In MS NAWM microglia, upregulated genes are involved in "lipid storage" and "lipid metabolism" (*EEPD1*, *LPL*, *PPARG*) or has been reported as early prognostic biomarker in MS (*CHI3L1*)[34]. Downregulated genes were *ADGRG1* and *MRC1*. Top DE genes in microglia from MS NAGM tissue were *SPP1*, *CXCR4*, and *GPNMB* involved in inflammatory responses. We thus found clear regional transcriptional differences for microglia in both control and MS donors.

**Co-expression networks for GM and WM microglia.** To determine transcriptional networks that are based on changes in patterns of gene expression rather than FCs of individual genes, we performed an unbiased weighted gene co-expression network analysis (WGCNA). This resulted in a network clustered into 21 modules (Fig. 3a). Each module was correlated to a module eigengene (ME). Region-related modules were represented by ME lightcyan (439 genes) and ME coral (3130 genes) for GM and by ME darkseagreen (71 genes) and ME black (3492 genes) for WM.

Several gene set enrichment analyses were performed for each module to determine genes significantly (false discovery rate-adjusted $p < 0.05$) overrepresented within Hallmarks, Gene ontology (GO) terms, Kyoto Encyclopedia of Genes and Genomes (KEGG) pathways, and diseases (Fig. 3b and Supplementary Data 3-6). Moreover, a connectivity analysis was performed to detect the genes with the highest intramodular connectivity (called hubgenes; Supplementary Data 7). The top hubgenes within each pathway and the heatmaps presenting total gene expression for modules of interest are displayed in Fig. 3c, d.

The module significantly correlated with control GM was ME lightcyan and showed enriched hallmark pathways for "complement" and "inflammatory response" and the GO terms "neutrophil mediated immunity" and "myeloid leukocyte mediated immunity" (Fig. 3b). The hubgenes *CR1* and *S100A12* are associated with the "complement" pathway and *MMP9*, together with *CR1* and *S100A12*, are related to the GO terms "neutrophil mediated immunity" and "myeloid leukocyte mediated immunity" in control GM microglia (Fig. 3c). The heatmap for module lightcyan showed interdonor variation in gene expression in the GM control group (Fig. 3d). ME coral was highly correlated with GM microglia in both control and MS tissue, and little variation in co-expressed genes for the GM group was found between donors (Fig. 3d). Hallmark enrichments within the coral module were "interferon gamma response" and "interferon alpha response," corresponding with the highly enriched GO terms "type I interferon signaling pathway" and "cellular response to type I interferon" (Fig. 3b). *STAT2*, a hubgene within ME coral is a transcription factor that regulates the interferon (IFN) response (Fig. 3c). Genes that are regulated downstream of the IFN response, like *IFI44* and *CCL2*, are

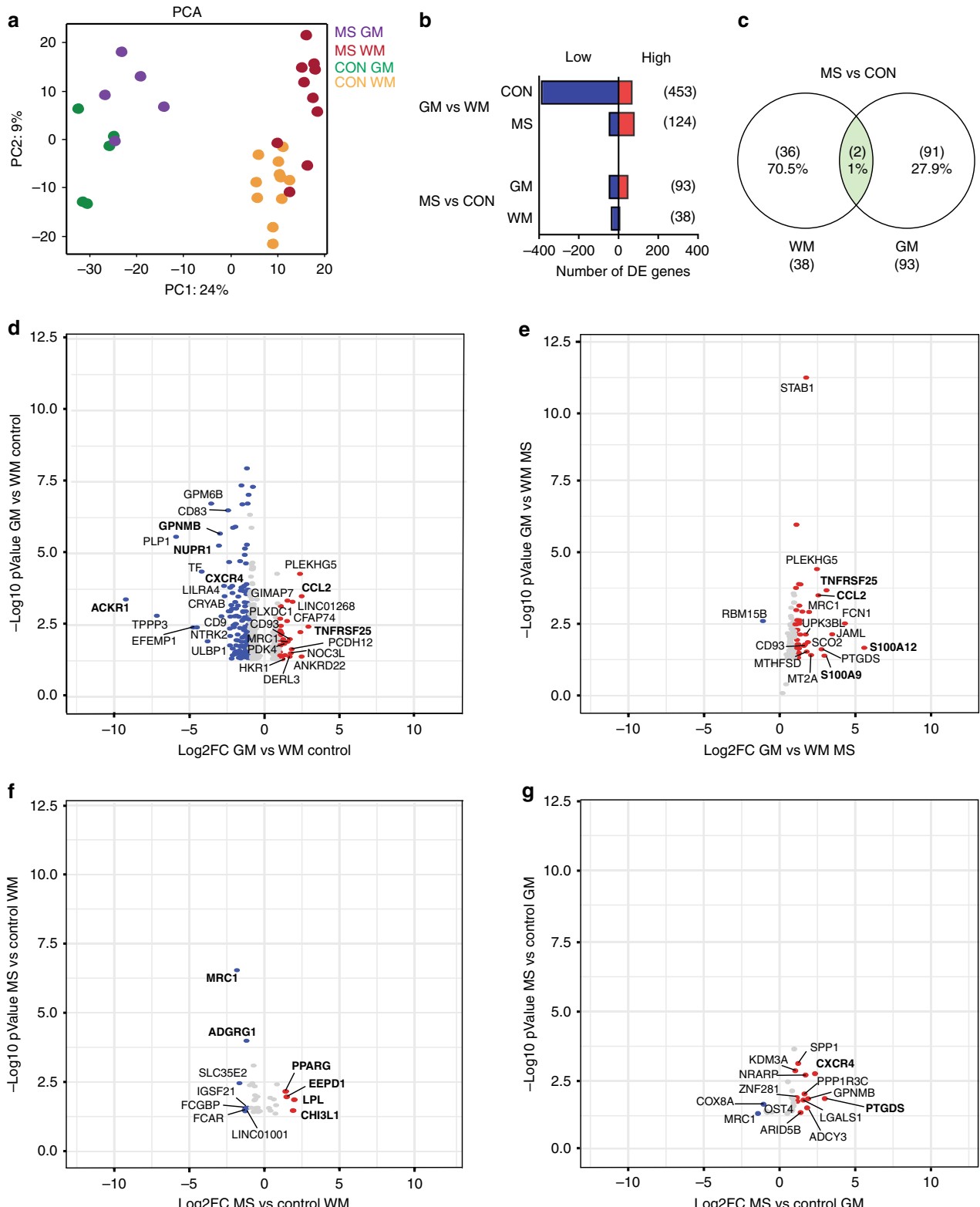

**Fig. 2** Regional and multiple sclerosis (MS)-specific changes in human microglia determined by differentially expressed (DE) genes. **a** Principal component analysis of global gene expression demonstrates profound differences between microglia from GM and WM and smaller differences between microglia from control and MS donors. **b** Number of DE genes for GM vs WM and MS vs CON, based on a fold change of > 2 or <−2 and an adjusted *p* value (false discovery rate <0.05). **c** A Venn diagram shows hardly any overlap between GM and WM MS microglia, based on DE genes. The two overlapping genes are *MRC1* and *LOC101927481*. **d**–**g** Volcano plots illustrating the top regulated genes in GM vs WM microglia from control (**d**) or MS (**e**) and in MS vs control microglia in WM (**f**) and GM (**g**) regions. Red and blue dots indicate positive or negative fold change, respectively

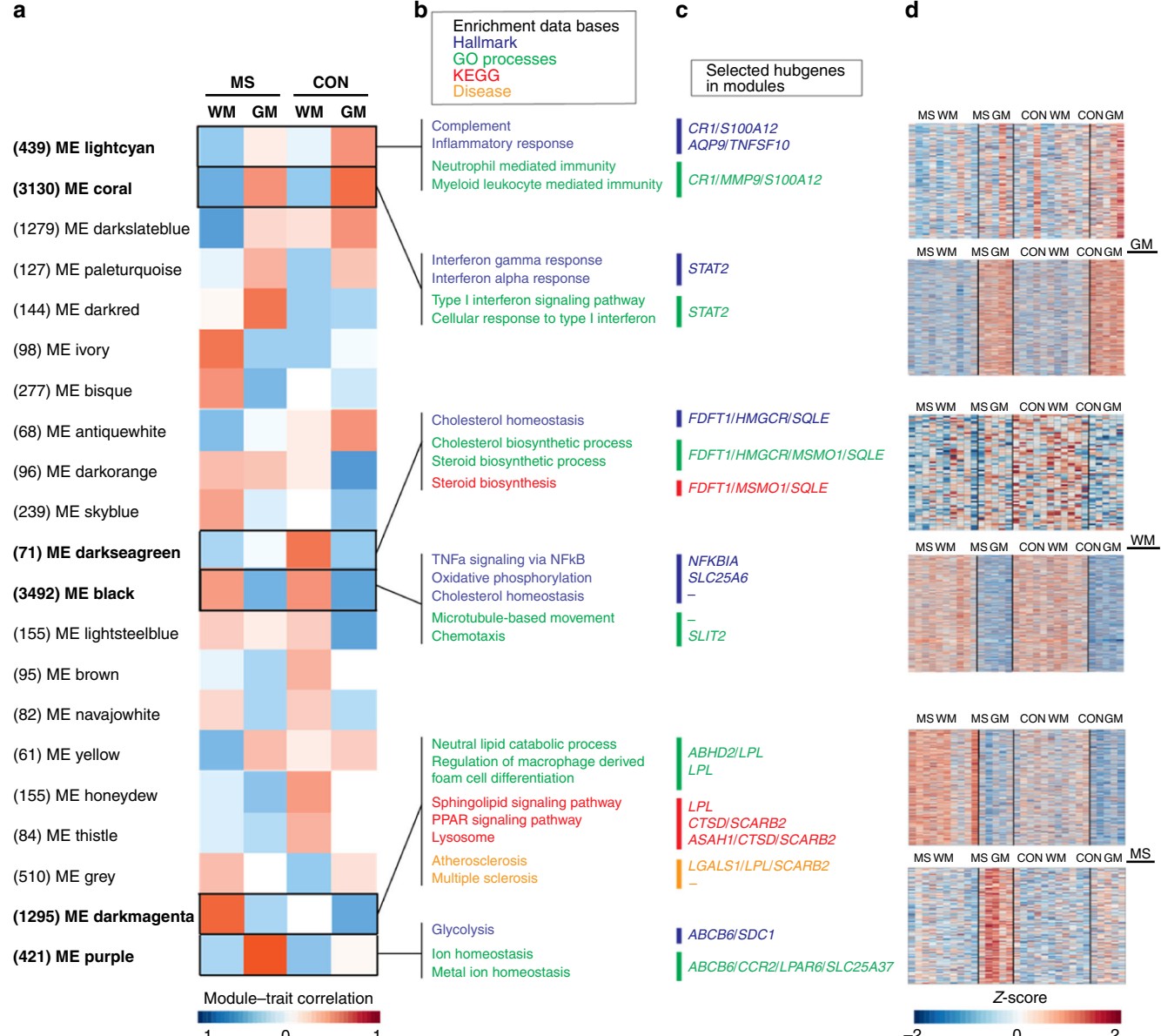

**Fig. 3** Co-expression networks for microglia in grey and white matter (GM and WM, respectively) and in normal-appearing multiple sclerosis (MS) tissue. **a** Twenty-one modules identified by weighted gene co-expression network analysis are provided with their module–trait correlation and the amount of genes that belong to each module displayed between brackets. **b, c** Modules of interest for GM and WM regions and for MS GM or WM tissue are highlighted with top enriched hallmarks, Gene ontology terms, Kyoto Encyclopedia of Genes and Genomes pathways, and disease terms as well as top hubgenes with the highest connectivity provided. Module eigengenes (MEs) lightcyan and coral correlate to GM in control or in both control and MS microglia, respectively. MEs darkseagreen and black correlate to WM in control microglia or in both control and MS, respectively. MEs darkmagenta and purple correlate to MS WM or GM, respectively. **d** Heatmaps display the expression of each gene in modules of interest, based on Z-score

significantly higher expressed in cortical microglia. In addition, protein expression of signal transducer and activator of transcription factor 2 (STAT2) by microglia in both GM and WM regions was confirmed by immunohistochemistry in tissue of control donors (Supplementary Figure 6).

ME darkseagreen represents control WM microglia with enriched hallmark pathways "cholesterol homeostasis" and "cholesterol and steroid biosynthetic process" and the associated KEGG pathway "steroid biosynthesis" (Fig. 3b). Hubgenes with high connectivity were *FDFT1*, *MSMO1*, *HMGCR*, and *SQLE* (Fig. 3c). High interdonor variation in gene expression patterns for WM control microglia was observed (Fig. 3d). Within ME black, corresponding to WM microglia in both control and MS tissue, top enriched hallmark pathways were "TNFα signaling

via NF-κB," "oxidative phosphorylation," and "cholesterol homeostasis." Enriched GO terms were "microtubule based movement" and "chemotaxis" (Fig. 3b). Hardly any variation in gene expression between donors for WM microglia was observed (Fig. 3d). The hubgene related to the "oxidative phosphorylation" pathway was *SLC25A6*, and *SLIT2* was related to the GO term "chemotaxis" (Fig. 3c). *NFKBIA* was identified as hubgene for the pathway "TNFα signaling via NF-κB" and was significantly higher expressed in WM microglia, together with increased expression of other nuclear factor (NF)-κB inhibitors and downstream targets of the NF-κB pathway, *IL8* and *PTGS2*.

NF-κB signaling can be activated via the canonical and the noncanonical pathway[35] (Supplementary Figure 7A). We determined the cellular localization of the NF-κB subunits

p65 (canonical pathway) and RelB (noncanonical pathway) in WM and GM microglia using immunofluorescence confocal microscopy in brain tissue of control donors. The majority of GM and WM microglia showed no nuclear translocation of these subunits (Supplementary Figure 7B-E). Moreover, western blot analysis of isolated microglia from control donors revealed no phosphorylation of IκBα (indicative of active canonical NF-κB signaling) or processing of p100 into p52 (indicative of active noncanonical NF-κB signaling) (Supplementary Figure 7F).

In conclusion, co-expression network analysis identified a clear difference in the immune regulatory profile of GM and WM microglia, as GM microglia showed a high expression of genes associated with type-I IFN response, and genes belonging to the NF-κB pathway were higher expressed in WM microglia. We observed no signs of NF-κB pathway activity in control GM and WM microglia.

**Co-expression networks for normal-appearing MS microglia.** Modules significantly correlated with MS were ME darkmagenta, related to microglia from NAWM, and ME purple for microglia from NAGM (Fig. 3a). We noticed that not one module correlated with MS-related changes in both GM and WM microglia, indicating that the MS-associated gene expression profile is region specific for microglia. In ME darkmagenta, overrepresented genes were related to GO terms "neutral lipid catabolic process" and "regulation of macrophage derived foam cell differentiation." KEGG pathways included "sphingolipid signaling pathway," "PPAR signaling pathway," and "lysosome" (Fig. 3b). Strikingly, disease ontology enrichment analysis showed "atherosclerosis and multiple sclerosis," including *LPL* as hubgene, involved in lipid metabolism. Several additional hubgenes within this module were related to the lysosomal compartment, *ASAH1* and *CTSD* have enzymatic functions, and *SCARB2* is known to be a lysosomal membrane receptor (Fig. 3c). Supplementary Figure 8 provides an overview of the genes within the KEGG pathway "lysosome." Enrichment analysis identified hallmark pathway "glycolysis" and the top GO terms "ion homeostasis" and "metal ion homeostasis" in ME purple (Fig. 3b). Hubgenes in this module were *ABCB6*, *CCR2*, *LPAR6*, *SDC1*, and *SLC25A37*, associated with metal ion homeostasis and glycolysis (Fig. 3c). Heatmaps for co-expressed genes showed some variation between MS donors within modules darkmagenta and purple, with three MS donors that showed a more similar expression pattern as in control donors for WM microglia, indicating heterogeneity of MS-related changes between donors (Fig. 3d).

To conclude, co-expression network analysis identified metabolic changes in microglia from normal-appearing MS tissue. We report co-expression changes of genes associated with glycolysis and metal ion homeostasis in NAGM MS microglia, pointing to changes in microglial metabolic state and a response to iron in the NAGM. In contrast, microglia in MS NAWM showed a higher co-expression of genes associated with the lysosomal pathway, lipid catabolism, and foam cell differentiation, suggesting early lipid processing.

**Conserved homeostatic transcriptome in NAWM-NAGM microglia.** In mouse models of neurodegenerative disorders, microglia downregulate the expression of homeostatic genes, including *P2RY12*, *P2RY13*, and *ADGRG1*[15]. Downregulation of *P2RY12* has also been described in normal-appearing MS tissue[36]. We identified hardly any changes in the expression of these homeostatic genes in normal-appearing MS tissue (Fig. 4a, b). Expression of the top 25 homeostatic signature genes, except for *USP2* in NAGM and *ADGRG1* in NAWM, was not affected in MS (Fig. 4c, d). Confirming the significantly lower gene expression of *ADGRG1* in MS, GPR56 protein expression was reduced in microglia isolated from MS NAWM compared to non-MS WM

microglia ($p = 0.003$; Fig. 4e). Our data indicate conservation of the microglial homeostatic state in MS normal-appearing tissue seemingly devoid of MS pathology.

**NAWM gene expression differences relate to MS lesion changes.** Microglia in MS NAWM showed significantly increased expression of *LPL*, *EEPD1*, and *CHI3L1* mRNA (Fig. 2f), which are implicated in lipid metabolism and therefore possibly involved in demyelination. To investigate whether these changes in MS NAWM indeed relate to MS lesion pathology, we also analyzed the expression of *LPL*, *EEPD1*, and *CHI3L1* in mixed active/inactive MS lesions. Both *LPL* ($p = 0.02$) and *CHI3L1* ($p = 0.003$) were significantly increased in these MS lesions as compared to NAWM tissue, whereas *EEPD1* expression was unaltered (Fig. 5a). Moreover, expression of *ADGRG1* was significantly reduced in NAWM microglia (Fig. 4d) but did not differ between NAWM tissue and MS lesions (Fig. 5a).

To specifically assess the expression of these genes in MS lesion-associated microglia, we analyzed the nuclear RNA expression from IRF8[+] nuclei from the same NAWM and lesion blocks we used to analyze whole-tissue gene expression. Interferon response factor 8 (IRF8) is a nuclear transcription factor highly enriched in murine microglia compared to other CNS cells[37] and bone marrow-derived myeloid cells[38]. To assess whether the IRF8 enrichment is conserved in human microglia, we demonstrated that IRF8[+] nuclei from WM indeed show enhanced expression of the microglia genes *CX3CR1* (FC 4.1, $p = 0.006$), *P2RY12* (FC 3.6, $p = 0.03$), and *CSF1R* (FC 7.4, $p = 0.02$) when compared to the whole nuclear fraction from WM tissue (Supplementary Figure 9). IRF8[+] nuclei isolated from MS lesions showed a significantly higher expression of *CHI3L1* ($p = 0.003$) and *LPL* ($p = 0.03$) when compared to IRF8[+] nuclei from NAWM MS tissue. Thus the increased expression of lipid metabolism genes in MS lesion tissue is conserved in the microglial fraction of the same lesions (Fig. 5b). Furthermore, expression of *EEPD1* and *ADGRG1* was not changed in IRF8[+] nuclei from lesions compared to NAWM (Fig. 5b).

## Discussion
The role of microglia in lesion initiation in both WM and GM regions in MS is not clear. To identify the transcriptome of microglia and possible early changes related to MS pathology, we studied normal-appearing tissue and analyzed the gene expression profile of microglia isolated from post-mortem GM and WM tissue. We identified transcriptional changes in microglia associated with known MS pathology, like iron metabolism in GM and lipid processing in WM. In addition, we confirmed that NAWM transcriptional changes are early signs of MS lesion pathology by showing similar upregulation in microglia nuclei isolated from active MS lesions. At the same time, the homeostatic profile of microglia was maintained in normal-appearing MS tissue, and the expression of genes, such as *P2RY12* and *TMEM119*, remained unchanged. In addition, we identified a clear region-specific profile for microglia in both non-neurological control and MS donors, which converge on immune regulatory pathways.

Our analysis showed a clear overlap of highly expressed genes in both GM and WM with the microglia transcriptomic signature recently identified in mice[28,29] and confirmed in humans[10,11]. In addition, we found several of these genes to be enriched in microglia, compared to autologous choroid plexus macrophages. Interestingly, the adhesion family GPCR GPR56 (*ADGRG1*) was distinctly expressed in both GM and WM microglia, compared to choroid plexus macrophages. This finding is in line with recent

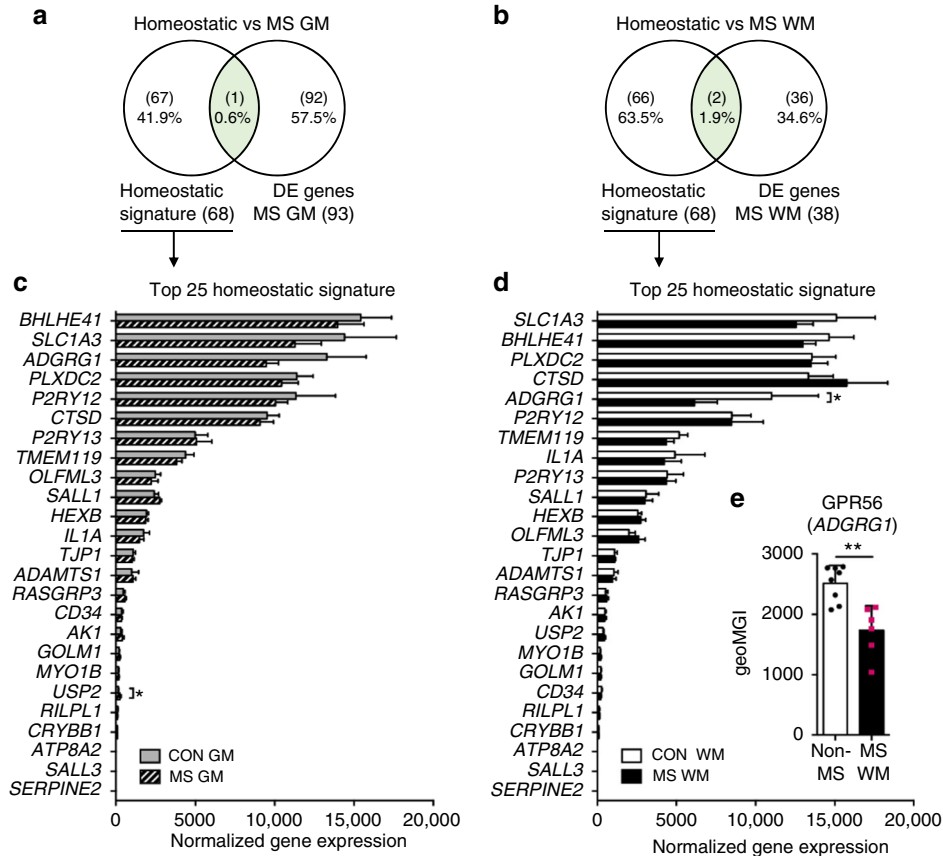

**Fig. 4** Conservation of homeostatic gene expression in microglia from normal-appearing multiple sclerosis (MS) tissue. **a**, **b** Venn diagrams show hardly any overlap between homeostatic genes identified in mouse microglia[15] and differentially expressed genes for grey matter (GM; $n = 5$) and white matter (WM; $n = 10$) MS microglia. **c**, **d** Expression of 25 homeostatic signature genes is amply changed in human microglia from both MS GM and WM, except for USP2 upregulation in GM and ADGRG1 downregulation in WM MS microglia. Two-way ANOVA: * adjusted $p < 0.05$. **e** Flow cytometric analysis confirms downregulation of GPR56 protein expression in MS WM microglia ($n = 6$) compared to non-MS WM microglia ($n = 8$). Mann–Whitney $U$ test: ** $p < 0.01$. Bars show mean with standard deviation

reported expression of *Adgrg1* in yolk sac-derived microglia, but not in bone marrow-derived microglia-like cells in mice, suggesting that GPR56 is a distinctive marker that allows to distinguish microglia from CNS-infiltrating macrophages under pathological conditions[31]. GPR56 is also highly expressed in cytotoxic lymphocytes, where it inhibits effector functions and keeps the cells in a quiescent state[39]. The function of GPR56 in microglia is not known yet, but may revolve around the maintenance of a homeostatic state.

The transcriptome of human microglia showed clear differences between WM and GM tissue. The tissue environment in the cortical region consists of neuronal cell bodies and low myelin density, which constitutes a substantially distinct environment for microglia as compared to WM regions that are rich in lipids and devoid of neuronal ligands that control microglia activity, such as CX3CL1, CD47, and CD200[40]. Besides an inherently different transcriptomic profile, cellular numbers could further dictate differences in microglial response to environmental changes. A higher microglia density, correlating with a higher yield of microglia isolated from post-mortem tissue, has been reported for WM regions[26,41]. Microglia in the cortex of non-neurological control donors showed a higher expression of genes related to the complement pathway, in particular high expression of the hub-gene complement receptor 1 (*CR1*). Co-expression of complement pathway genes might relate to an essential role for microglia in complement-mediated synapse pruning under homeostatic conditions in the cortex[42,43]. In control and MS donors, cortical

microglia co-expressed genes involved in the IFN type-I response. This response is important in limiting viral spread in the CNS and is produced by many cell types, including neurons and microglia[44–46]. Our dataset showed a clear difference in expression of type-I IFN genes between GM and WM microglia under non-inflammatory conditions. Specifically, we observed higher expression of the IFN genes *STAT2* and *IRF9* in cortical microglia, which can form a complex that translocates to the nucleus and modulates the type-I IFN response[44]. High expression of type-I IFN genes might prime cortical microglia to quickly respond to viral infections and preventing neuronal damage.

Higher expression of genes belonging to the NF-κB pathway was found in WM microglia in both control and normal-appearing MS tissue. NF-κB is well known for its role in inflammatory responses and is an essential player in MS pathology[47]. We here studied non-inflammatory conditions and identified higher expression of NF-κB inhibitor genes and downstream targets *IL8* and *PTGS2* in WM microglia, compared to GM microglia. Suppression of NF-κB pathway activation under non-inflammatory conditions might be essential to keep microglia in a quiescent state and prevent full activation. Indeed, we found no signs of NF-κB pathway activation in GM and WM microglia. Presumably, higher expression of downstream transcripts *IL8* and *PTGS2* in WM microglia may be related to other signaling pathways. Our results on microglial GM and WM differences are supported by a recent transcriptomic study that reports a region-specific profile of murine microglia by

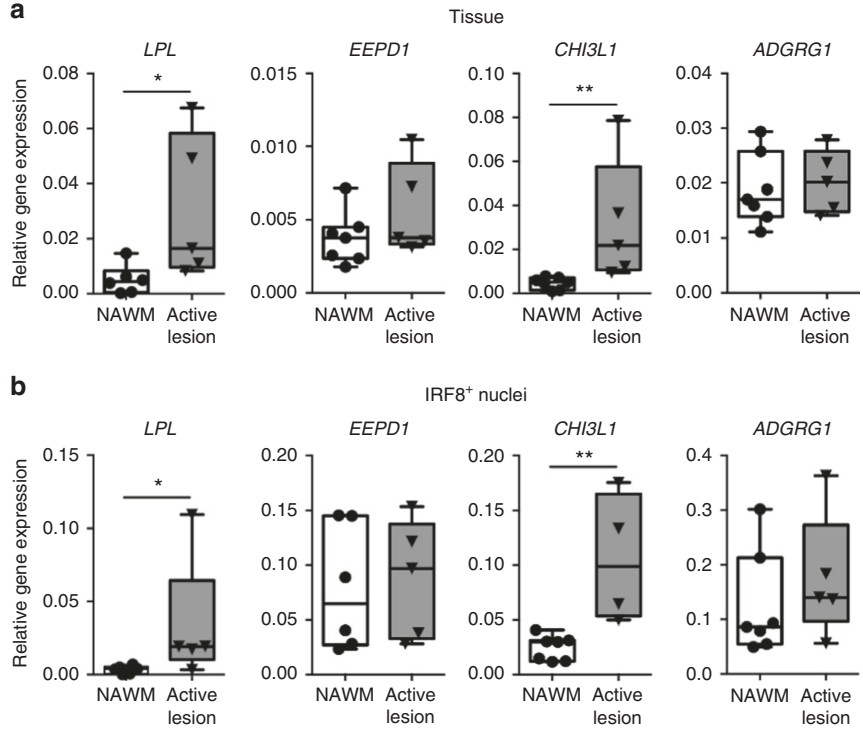

**Fig. 5** Gene expression differences observed in normal-appearing white matter (NAWM) microglia relate to changes found in multiple sclerosis (MS) lesions. **a** Excised mixed active/inactive MS lesion tissue ($n = 5$) shows increased expression of the differentially expressed genes *LPL* and *CHI3L1* compared to NAWM ($n = 7$) determined by quantitative reverse transcription polymerase chain reaction (RT-qPCR). *EEPD1* and *ADGRG1* expression is unaltered. Mann–Whitney *U* test: *$p < 0.05$, **$p < 0.01$. **b** IRF8+ nuclei isolated from the same mixed active/inactive MS lesions demonstrates increased expression of *CHI3L1* and *LPL*, compared to IRF8+ nuclei from NAWM, determined by RT-qPCR. In line with the tissue data, the expression of *EEPD1* and *ADGRG1* is not changed in IRF8+ nuclei from MS lesions. Mann–Whitney *U* one-tailed test: *$p < 0.05$, **$p < 0.01$. Box plot center line shows median; whiskers show minimum–maximum values

differences in immune regulatory pathways, like type-I IFN response, between different brain regions[48]. The regional microglial differences in immune regulatory pathways are of high importance to understand their role in GM or WM MS lesion initiation. GM lesions are characterized by lower numbers of infiltrating leukocytes[3,4], which can be explained by the high expression of IFN genes by cortical microglia to prevent leukocyte infiltration in GM tissue, which was demonstrated by Khorooshi and Owens[49].

Interestingly, DE genes that show higher expression in WM compared to GM control microglia and are associated with the NF-κB pathway, like *NFKBIZ* and *NFKBIA*, are not differentially expressed between GM and WM regions in MS anymore. Moreover, the number of DE genes between GM and WM regions was remarkably lower in microglia isolated from MS donors, indicating that microglia in MS may start losing their region-specific profile. However, we found that, in NAGM and NAWM MS tissue, expression of homeostatic genes in microglia is preserved, except for *ADGRG1* in NAWM and *USP2* in NAGM, suggesting a quiescent basal state of microglia in normal-appearing MS tissue. Our data are in contrast to those of Zrzavy and colleagues, who report downregulation of *P2RY12* and *TMEM119* and upregulation of *ADGRG1* already in normal-appearing MS tissue[36]. This discrepancy could be due to the fact that normal-appearing regions in this study were dissected from tissue situated adjacent to active lesions of patients with acute MS, who died within 4 months after disease onset, suggesting more severe MS pathology than in our cases.

In addition to the maintained homeostatic signature in normal-appearing MS tissue, we report clear MS-associated

metabolic changes in both NAGM and NAWM microglia. We report lipid processing as a major pathway that was altered in microglia from NAWM MS tissue, together with the lysosomal and peroxisome proliferator-activated receptor (PPAR) signaling pathways. The endonuclease-exonuclease-phosphatase family domain containing 1 (*EEPD1*) gene showed an increased expression in NAWM MS microglia and was recently identified as new liver X-receptor target to promote cholesterol efflux in macrophages[50]. In addition, both lipoprotein lipase (*LPL*) and *PPARG* were upregulated and are well-studied genes in the atherosclerotic field. *LPL* expression is regulated by the transcriptional factor PPARγ and mediates lipid processing and cholesterol efflux[51]. Both *LPL* and *PPARG* play a role in foam cell formation in the atherosclerotic plaque[52].

To study whether these lipid metabolism changes observed in NAWM MS microglia relate to MS pathology, we determined the expression of NAWM DE genes in MS lesions and showed that *LPL* and *CHI3L1* expression was indeed increased in MS lesions as compared to NAWM tissue. In order to assess transcript levels specifically in microglia, we developed a sorting strategy to enrich for microglial nuclei, based on the expression of IRF8. The transcriptional changes in whole tissue were conserved in the nuclear IRF8+ fraction, isolated from the same MS lesions and NAWM tissue. In addition, expression of *ADGRG1* was not changed in MS lesions, demonstrating that reduced expression of *ADGRG1* in NAWM is not sustained in demyelinating MS lesions and indicating that *ADGRG1* might be a useful marker to distinguish microglia from infiltrating macrophages in active MS lesions. However, more research on *ADGRG1*/GPR56 under pathological conditions is needed to define the stable expression

on microglia under conditions where microglia and infiltrating macrophages are both present. The MS lesions selected for this study contained many active foamy microglia/macrophages in the rim, indicating presence of demyelinating MS pathology. Therefore, the high expression of lipid metabolism genes by these foamy cells demonstrates that the microglial transcriptional changes we already observed in NAWM are likely early signs of MS lesion pathology.

The upregulation of genes involved in lipid metabolism, lysosomal function, and foam cell formation in NAWM tissue illustrates that microglia undergo changes due to lipid uptake and processing in tissue without radiological signs of demyelination. Interestingly, we previously reported that microglia isolated from control donors more efficiently phagocytose MS myelin as compared to myelin isolated from healthy control brain donors, and a study by Wheeler and colleagues showed altered lipid composition of myelin in normal-appearing MS tissue[24,25]. Changes in myelin lipid composition in NAWM in MS may go unnoticed in conventional immunohistochemistry studies. However, such small changes may have far-reaching consequences when microglia interact with and phagocytose myelinated axons. Another sign that supports a microglial response in NAWM is increased CHI3L1 expression, as CHI3L1 has been identified as early prognostic biomarker in MS and plays a role in limiting inflammation[34,53]. Other causes of possible changes in microglia–myelin interaction in NAWM in MS are related to complement opsonization[18,54] or Wallerian degeneration[55] in response to MS lesions situated close by, which might attract microglia for targeted myelin removal and contributes to lesion formation.

In contrast, microglia in NAGM MS tissue showed a different transcriptional profile associated with glycolysis and iron homeostasis. Iron accumulation is well described in the brains of MS patients, and accumulation of iron in microglia is a pathological hallmark of MS[56]. An MRI study showed iron depositions in deep GM regions in patients with clinical isolated syndrome, suggesting iron depositions as early hallmark of MS[57]. The observed increased expression of iron metabolism genes in cortical MS microglia indicates a response to iron depositions in the cortex. More specifically, we found increased expression of the mitochondrial importer mittoferin-1 (SLC25A37) and the mitochondrial transporter ABCB6 in MS NAGM microglia, both involved in maintaining cellular iron homeostasis[58]. Disturbed iron homeostasis could be a result of increased iron-rich oligodendrocyte debris in the environment[59]. Besides disturbed iron homeostasis, increased glycolysis was observed in NAGM MS microglia and might be due to iron accumulation, which has been shown in mouse microglia to increase the glycolytic response[60].

We here report the first signs of iron or lipid uptake by microglia in normal-appearing MS tissue, resulting in metabolic changes, but not affecting homeostatic signature genes. The downstream effect of increased iron uptake by microglia has to be studied in more detail, but the detection of increased glycolytic genes may indicate that microglia require a higher energy level for iron metabolism. In addition, long-term iron uptake, resulting in iron accumulation in microglia, might trigger the production of reactive oxygen species, dysfunctional mitochondria, and microglial dysfunction[59]. In line herewith, increased expression of the DE genes CXCR4, GPNMB, and SPP1 in NAGM suggests that early microglial changes relate to a neurodegenerative microglia profile, as upregulation of these genes was earlier observed in experimental mouse models for MS and Alzheimer's disease (AD)[15,16]. Therefore, these microglial changes in NAGM tissue might be early signs of pathology.

In microglia from NAWM tissue, early signs of lipid processing and PPARγ signaling might induce an anti-inflammatory phenotype, as lipid processing controls the inflammatory response and involves PPARγ signaling[61,62]. PPARγ agonists are used to treat patients with atherosclerosis, and several studies report the beneficial effect of PPARγ agonists as therapeutic target for MS[63,64], providing evidence for a beneficial role of microglia in NAWM. As microglia constantly survey their environment to maintain brain homeostasis, which includes the targeted removal of apoptotic oligodendrocytes and myelin debris[14], myelin clearance by microglia does not change their homeostatic profile and raises the question if this will result in MS lesion initiation. Importantly, in vivo studies in animal models for MS with disrupted microglial functions have shown that myelin debris clearance is crucial for remyelination and repair[5]. Based on our findings, we propose that microglia in normal-appearing tissue need a second hit to become fully activated, to play an initiating role in MS lesion formation[14].

In NAWM, clusters of activated microglia have been described that respond to axonal and myelin changes[54]. Together with the recent findings that only a small subset of microglia show a disease-related profile in an animal model for AD, based on single-cell sequencing[16], this illustrates that, also within normal-appearing MS tissue, a heterogeneous population of microglia can exist. The microglial changes that we observed in normal-appearing MS tissue are actually diluted by identifying the transcriptome of the bulk of microglial cells. Therefore, a single-cell approach or different cell purification methods are needed to study specifically these microglial clusters. Future studies should focus on studying these microglial clusters, as we hypothesize that these clustered microglia are initiators of lesion formation.

In summary, we identified a region-specific transcriptional profile for human microglia in GM and WM tissue, which is important for studying their role under healthy and neuroinflammatory conditions. Regional differences in the microglial response to different stimuli are especially important to consider in the search for novel therapeutic interventions. Furthermore, we identified microglial changes in normal-appearing MS tissue associated with known MS pathology, pointing toward an important role for microglial metabolic changes in MS pathology. Future studies should focus on microglia clusters and myelin composition in normal-appearing MS tissue to further identify molecular changes in normal-appearing MS tissue and to identify initial signs of MS lesion formation that may be of interest for therapeutic targeting.

## Methods

**Human post-mortem tissue**. Human brain cortical GM, corpus callosum WM, and choroid plexus tissue was provided by the Netherlands Brain Bank (NBB, Amsterdam, The Netherlands; https://www.brainbank.nl). Informed consent to perform autopsies and the use of tissue and clinical data for research purpose were obtained from donors and approved by the Ethical Committee of the VU University medical center (VUmc, Amsterdam, The Netherlands).

GM and WM tissue blocks from occipital cortex ($n = 5$) and corpus callosum ($n = 11$), respectively, of non-neurological control donors were collected. NAGM ($n = 5$) and NAWM ($n = 10$) tissue blocks of MS donors were dissected at autopsy on post-mortem MRI guidance[65]. Brain slices of 10 mm were analyzed by MRI to determine absence of MS lesions in occipital cortex and corpus callosum. A small part of the tissue blocks used for microglia isolation was cut off using a scalpel, snap-frozen in liquid nitrogen, and immunohistochemically stained for microglia activation markers CD68 (M0814; Dako, Glostrup, Denmark), HLA-DP/Q/R (HLA-DR, M0775; Dako)[66], and myelin PLP (MCA839G; AbD Serotec, Oxford, UK)[27] to study microglia morphology and myelin integrity.

Neurological diagnosis was confirmed post-mortem by a neuropathologist, based on both clinical and pathological data. Non-neurological control donors with cognitive problems, based on clinical data, were excluded from the analysis. Donor characteristics are displayed in Table 1 and Supplementary Data 1.

**Microglia isolation**. Corpus callosum and occipital cortex tissue blocks of 4–6 g were dissected at autopsy and stored in Hibernate A medium (Invitrogen, Carlsbad, CA, USA) at 4 °C until further processing. Microglia isolations were performed as described before[26]. Briefly, brain tissue was mechanically dissociated using a tissue

homogenizer (VWR, Radnor, PA, USA), followed by enzymatic digestion with collagenase (300 U/ml; Worthington, Lakewood, NY, USA) for 60 min or with trypsin (Invitrogen) for 45 min at 37 °C in Hibernate A medium supplemented with DNAse I (Roche, Basel, Switzerland). For flow cytometric analysis, enzymatic digestion was omitted. Percoll (GE Healthcare, Little Chalfont, UK) density centrifugation was performed, and microglia were collected from the interlayer. Magnetic-activated cell sorting (Miltenyi Biotec, Bergisch Gladbach, Germany) was performed for negative selection of neutrophils by magnetic anti-CD15 beads and positive selection of microglia by magnetic anti-CD11b beads (Miltenyi Biotec). Viable cells were counted, using a hemocytometer (Optic Labor, Friedrichsdorf, Germany), and stored in 1 ml cold TRIsure (Bioline, London, UK) at −80 °C for further analysis. To assess the purity of isolated cells, CD45, CD11b, and CD15 expression was analyzed by flow cytometry. Choroid plexus macrophages were isolated by a similar procedure, aside from using trypsin for enzymatic digestion and excluding the CD15 magnetic bead step.

**Isolation and sorting of IRF8⁺ nuclei**. Frozen tissue containing mixed active/inactive MS lesions ($n = 5$) and NAWM tissue ($n = 7$), matched for age, was provided by the NBB. Tissue was double stained with HLA-DR/PLP for lesion characterization, previously described by Luchetti and colleagues[27]. Tissue blocks of 5 donors contained a mixed active/inactive lesions (type 3.2 or 3.3), whereas NAWM blocks were devoid of demyelination. From each NAWM block, 11–13 50-μm sections were cut, one for whole tissue RNA, 10–12 for nuclei isolation, and the first and the last section of each block (10 μm) was used to determine microglia activation and myelin integrity using a HLA-DR/PLP double staining[27]. For tissue blocks containing lesions, we followed the same protocol with the exception that HLA-DR/PLP staining was used to guide dissection of lesions with a scalpel. This ensured that cryosectioning resulted in the collection of only lesioned WM.

We then processed the sections for use in nuclear sorting as described by Krishnaswami and colleagues[67]. Briefly, tissue sections were homogenized in 1 ml homogenization buffer (1 μM DTT (Thermo Fisher Scientific), 1× protease inhibitor (Roche), 80 U/ml RNAaseIn (Promega, Madison, WI, USA), and 1% Triton X-100) diluted in nuclei isolation medium #1 (NIM #1; 250 mM sucrose, 25 mM KCl, 5 mM MgCl₂, 10 mM Tris buffer pH8 diluted in nuclease-free water) and filtered through a 30-μm cell strainer. The number of nuclei was counted using a cell hemocytometer (Optic Labor). Nuclei were incubated for 1 h with Hoechst (1:1000; Invitrogen) and PE-labeled IRF8 antibody (clone U31–644, 1:50; BD Biosciences) in staining buffer (0.5% RNAse-free bovine serum albumin (BSA) and 0.2 U/μl RNaseIn in RNase phosphate-buffered saline (PBS) pH 7.4) with 1% normal human serum at 4 °C. Background staining was determined using an isotype control antibody (IgG1-PE, clone P3.6.2.8.1, 1:25; Invitrogen).

For nuclei sorting, a Sony SH800S cell sorter (Sony Biotechnology, San Jose, CA, USA) was used to detect Hoechst⁺ nuclei and sort IRF8⁺ and Hoechst⁺ fractions. Nuclei were immediately lysed in RNA lysis buffer (Qiagen, RNeasy Isolation Mini Kit).

**RNA isolation**. Isolation of RNA was performed according to the manufacturer's instructions, chloroform was added, and after centrifugation, the aqueous phase was collected. Subsequently, cold isopropanol and 1 μg glycogen (Roche) were added for RNA precipitation at −20 °C for 30 min. After centrifugation, the pellet was washed twice with 75% cold ethanol, and RNA was dissolved in 10 μl deionized water. For donor 11–008, whole tissue and sorted nuclei from frozen tissue, the Qiagen RNeasy Isolation Mini Kit (Qiagen, Hilden, Germany) was used to isolate RNA according to the manufacturer's instructions. Briefly, after phase separation, the aqueous phase was transferred to a Qiagen mini column and eluated in 10 μl deionized water.

**Sample preparation and RNA sequencing**. The NEBNext Ultra Directional RNA Library Prep Kit from Illumina (San Diego, CA, USA) was used to process 38 samples (22 WM and 16 GM) (GenomeScan, Leiden, The Netherlands). The sample preparation was performed according to the protocol of NEBNext Ultra Directional RNA Library Prep Kit from Illumina (NEB #E7420). Briefly, mRNA was isolated from total RNA using the oligo-dT magnetic beads. After fragmentation of the mRNA, a cDNA synthesis was performed with an input of 50–100 ng RNA. This cDNA was used for ligation with the sequencing adapters and PCR amplification of the resulting product. The quality (RIN) and yield after sample preparation was measured with the Fragment Analyzer. Thirty-one samples (21 WM and 10 GM) met the quality criteria and were selected for sequencing. The size of the resulting products was consistent with the expected size distribution (a broad peak between 400 and 700 bp). Clustering and DNA sequencing using the Illumina NextSeq500 SR75 was performed according to the manufacturer's protocols. A concentration of 1.5 pM of DNA was used. At least 1.9 Gb (25 million SR75 reads) were generated per sample with a quality score of ≥30. NextSeq control software v1/4/8 was used. Image analysis, base calling, and quality check was performed with the Illumina data analysis pipeline RTA v2.4.6 and Bcl2fastq v2.17.

**Data preprocessing and RNA sequencing analysis**. Details on workflow for RNA sequencing analysis are displayed in Supplementary Figure 1. After base calling and de-multiplexing using CASAVA version 1.8 (Illumina), the 75 bp

single-end reads were aligned to the human reference genome hg19 from UCSC (https://genome.ucsc.edu) by HISAT version 0.1.7-beta (http://www.ccb.jhu.edu/software/hisat), using the default parameters. After mapping of the reads to the genome, we imported the data into Partek Genomics Suite V6.6 (Partek, St. Louis, MO, USA) to quantify the number of reads mapped to each gene annotated in the RefSeq hg19 (GRCh37) annotation downloaded in May 2016. Raw read counts were imported into R and normalized using the Bioconductor package DESeq2 package[68] using default parameter. Subsequent to normalization, all transcripts having a maximum over all group means <100 were removed. After dismissing the low expressed transcripts, the data comprised of 12,318 present transcripts. Unwanted or hidden sources of variation, such as the technical variance introduced by donors, were modeled during the normalization or were removed using the surrogate variable analysis (sva) package[69]. The normalized rlog-transformed expression values were adjusted according to the surrogate variables identified by sva using the function "removeBatchEffect" from the limma package[70].

DE genes analysis: A two-way analysis of variance model was performed to calculate the DE genes between all groups. DE genes were defined by a FC of >2 or <−2 and an adjusted $p$ value <0.05. To visualize the structure within the data, we performed PCA on all present genes. Venn diagrams were created with the online software tool Venny (http://bioinfogp.cnb.csic.es/tools/venny). Vulcano plots were created to visualize the top DE genes with at least 500 counts for all four groups in R using ggplot2 version 2.2.1.

Co-expression network analysis: To determine gene clusters associated with GM, WM, MS, or control, we used the present genes, corrected by sva using 7 iterations, and applied the R implementation of the WGCNA. We performed WGCNA clustering using the "1-TOMsimilarityFromExpr" function with the network type "signed hybrid", a power parameter of 6, and a minimum module size of 20, dissecting the data into 21 modules. A WGCNA was performed to describe the co-expression patterns across all samples in an unbiased way and cluster genes with a similar expression pattern into modules. In addition, the module representative principal component (ME) was assigned to one of the four comparisons, which results in a correlation.

Hub genes: Hub genes are highly connected nodes, which are involved in much more interactions in the whole network and thus possibly be more important than low connected genes. Gene connectivity defines the connection strength of a gene connected to other genes in a global network. Therefore, one hub gene with the highest connectivity in the selected module was extracted using the "chooseTopHubInEachModule" function in the WGCNA package.

Gene set enrichment analysis: Hallmark, GO, KEGG, and disease[71] enrichment analysis on modules defined by WGCNA was performed with clusterProfiler[72] using all present genes as background. KEGG pathways were visualized using the R package pathway. The gene names were colored based on FCs of the respective comparison.

**DNA synthesis and quantitative RT-PCR**. For cDNA synthesis, the QuantiTect Reverse Transcription Kit (Qiagen) was used. For the detection of microglial genes after culturing, we obtained cDNA from primary human WM microglia. Microglia from 4 different donors were lysed acutely after isolation as well as cultured for 4 days to investigate culture-induced changes[26]. Purified 100 ng RNA from isolated microglia or macrophages, 5–12 ng nuclear RNA from IRF8⁺-sorted nuclei, or whole tissue RNA was incubated for 2 min at 42 °C with gDNA wipeout buffer, put on ice, and mixed with QuantiTect Buffer, RT Primer Mix, and Quantitect Reverse Transcriptase. After 30 min incubation at 42 °C, the samples were incubated at 95 °C for 3 min. An input of 3.5 ng cDNA was mixed with 5 μl SYBR Green PCR Master Mix (Applied Biosystems, Foster City, CA, USA) and 1.5 μl primer pairs to obtain a final volume of 10 μl. For nuclei, 0.5–1.2 ng cDNA was mixed with 17 μl SYBR Green PCR Master Mix and 2 μl primer pairs to obtain a final volume of 20 μl. Analysis was performed by ABI Prism 7300 Sequence Detection System (Applied Biosystems).

Primer pairs were designed using the online tool Integrated DNA Technologies (eu.itdna.com) using the following criteria: 50% GC content, same Tm, and min–max amplicon size of 80–120 base pairs. Primer pairs to detect RNA in sorted IRF8⁺ nuclei were designed to detect unspliced DNA. Primer specificity was tested on cDNA derived from pooled brain tissue of MS and control donors, and selection of optimal primers was based on dissociation curve, PCR product, and negative selection for primer dimers on 8% sodium dodecyl sulfate polyacrylamide gel electrophoresis (SDS-PAGE) gel (Supplementary Table 1). Target gene expression was normalized to the mean of glyceraldehyde 3-phosphate dehydrogenase (GAPDH) or elongation factor-1 alpha (EEF1A1), and fold differences were calculated using the $2^{-\Delta\Delta CT}$ method[73].

**Immunohistochemistry**. Frozen tissue sections (20 μm) of cortical region (containing both GM and WM) from control donors ($n = 3$) were fixated for 10 min in 4% paraformaldehyde. Sections were incubated overnight at 4 °C with primary antibodies for NF-κB subunits RelB (SC-226, 1:100; Santa Cruz, Dallas, TX, USA) or p65 (SC-372, 1:100; Santa Cruz) and HLA-DR/DQ/DP (M0775, 1:1000, Dako, Glostrup, Denmark) or for STAT2 (SC-166201, 1:100; Santa Cruz) and IBA1 (019–19741, 1:500; Wako, Osaka, Japan) diluted in incubation buffer (1% bovine serum albumin + 0.5% Triton X-100 in Tris-buffered saline, pH 7.6 (TBS)). Fluorescently labeled secondary antibody (711–165–152, Cy3-labeled, 1:400;

Jackson ImmunoResearch, West Grove, PA, USA) or biotinylated secondary antibody (BA-2001, 1:400; Vector Laboratories, Burlingame, CA, USA) were diluted in incubation buffer and incubated for 1 h at room temperature (RT), followed by incubation with streptavidin-Alexa488 antibody (061–540–084, 1:1200; Jackson ImmunoResearch) for 45 min. Sections were incubated for 10 min in Hoechst (1:1000) for nuclei staining and embedded in fluorescence mounting medium (DAKO). Immunoreactivity was examined using a confocal laser scanning microscope (SP8; Leica, Wetzlar, Germany) with the software LASX, using Z-stack acquisition (magnification ×63, step size 1 μm, zoom factor 2.5). Pictures were processed and analyzed using Fiji plugin for ImageJ1 software.

**Western blot analysis**. Isolated microglia ($n = 3$) were lysed in ice-cold 10× RIPA Buffer (Cell Signaling Technology, Danvers, MA, USA). Protein samples were subjected to SDS-PAGE and transferred to polyvinylidene difluoride membranes (Millipore, Bedford, MA, USA). For GPR56 detection, blotted membranes were incubated in blocking buffer (5% BSA in TBS with 0.05% Tween-20 (TBS-T)) for 1 h, followed by 2 h incubation at RT for primary antibody in blocking buffer. Following extensive washes in TBS-T, membranes were incubated with horseradish peroxidase (HRP)-conjugated secondary antibody for 1 h at RT (Dako) (1:5000 in blocking buffer). Membranes were extensively washed, and the bound HRP signal was detected by chemiluminescence for 5 min (Amersham ECL; GE Healthcare or Supersignal West Pico PLUS; Thermo Fisher Scientific, Waltham, MA, USA) using the software Image Quant LAS 4000 mini version 1.3.

For the detection of NF-κB subunits, blotted membranes were incubated in blocking buffer (5% blotting-grade blocker (Bio-Rad, Hercules, USA) in TBS-T) for 1 h, followed by overnight incubation with the NF-κB antibodies in 2.5% blocking buffer at 4 °C. Following extensive washes in TBS-T, membranes were incubated with HRP-conjugated secondary antibody for 1 h at RT (Dako) (1:2000 in 2.5% blocking buffer). Membranes were extensively washed, and the bound HRP signal was detected by chemiluminescence using Lumi-Light Western blotting substrate (Roche), using the software Image Quant LAS 4000 mini version 1.3. Primary antibodies are displayed in Supplementary Table 2. Unprocessed blots are shown in Supplementary Data 8 and 9.

**Flow cytometric analysis**. Extracellular protein expression on isolated human WM microglia ($n = 3$) and choroid plexus macrophages ($n = 3$) was detected by flow cytometry. After blocking unwanted binding of antibodies to Fc receptor with human FcR Blocking Reagent (Miltenyi Biotec), microglia and macrophages were incubated for 30 min with conjugated antibodies (Supplementary Table 3) in beads buffer (0.5% BSA+2 mM EDTA in PBS, pH 7.6) at 4 °C. For detection of viable cells, fixable Viability Dye eFluor$^{TM}$ 506 (1:500; Thermo Fisher Scientific) was added. Background staining was determined using fluorescence minus one controls. Protein expression was measured on a 3-laser BD FACSCanto II$^{TM}$ machine (BD Biosciences, San Diego, CA, USA) with the software BD DIVA version 8.1, and median fluorescence intensity was determined with the FlowJo software version 10.1 (Ashland, OR, USA).

**Statistical analysis**. Statistical analysis of RT-qPCR, flow cytometry, and western blot experiments was performed using the GraphPad Prism version 7.01 software (GraphPad Inc., La Jolla, CA, USA). After testing for normality (Shapiro–Wilk normality test), either parametric or non-parametric tests were used to define significance. Statistical testing for each experiment is indicated in the figure legends. $p$ Values <0.05 were considered significant.

**Reporting summary**. Further information on experimental design is available in the Nature Research Reporting Summary linked to this article.

## Data availability
The RNA-sequencing dataset is available online in the Gene Expression Omnibus (GEO) database (https://www.ncbi.nlm.nih.gov). The GEO accession number is GSE111972.

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

## Acknowledgements

The authors acknowledge the Netherlands Brain Bank (https://www.brainbank.nl) for providing the donor material and A. Konstantoulea for performing RT-qPCR experiments. This study was supported by the Dutch MS Research Foundation (grant 13–830 MS), the Thyssen Foundation (2015–00387), and the German Research Foundation (FOR 2149).

## Author contributions

M.v.d.P isolated human microglia, analyzed gene and protein expression, carried out RNA sequencing data analysis, and drafted the manuscript. T.U. analyzed the RNA sequencing data. M.R.M. developed methods, interpreted data, and drafted the manuscript. C.-C.H. performed protein expression analyses. S.S.M.M. organized RNA sequencing analysis. A.A. isolated RNA from tissue and IRF8+ nuclei. K.G.S. isolated human microglia. B.H. performed western blot analysis. S.W.T. interpreted data and drafted parts of the manuscript. J.L.S. supervised the RNA sequencing data analysis. J.H. and I.H. were responsible for study design and coordination, data interpretation, and drafting of the manuscript. All authors read and approved the final manuscript.

## Additional information

**Competing interests:** The authors declare no competing interests.

**Journal Peer Review Information:** *Nature Communications* thanks Oleg Butovsky and other anonymous reviewer(s) for their contribution to the peer review of this work. Peer reviewer reports are available.

