## [Peer Review File · Nature Communications]

Reviewers' comments:

Reviewer #2 (Remarks to the Author):

Overall an interesting manuscript that is logically and well-written. The findings will serve an important resource for human microglia transcriptome in early stages of MS. However, there are several comments to be addressed before acceptance.

Major Comments:

1) There is no gene expression profiling of microglia from MS lesions. Thus, the manuscript lacks a comparison between the microglia from normal appearing GM/WM and microglia within the lesions.

2) The authors should provide additional information on how normal-appearing GM and WM were defined (all they say is by using MRI guidance). How far away were selected areas (corpus callosum and occipital cortex) from pre-existing MS lesions? And why did authors choose occipital cortex in particular? Temporal lobes are more commonly affected in MS, and therefore may have been a better site to look for "pre-MS" microglial transcriptional changes. When did these patients take MRI? Which sequences did the authors use? According to line 270, "due to MRI, had a greater post-mortem delay", it seems that the patients took MRI after they passed away. Lacking the supply of blood and oxygen, will the brain change drastically in MRI? Are the entire corpus callosum or occipital gyrus normal appearing in the MRI? If not, how could the authors get the non-lesion part of these regions? Please clarify it.

1) Why the authors focus on Gpr56? The authors should provide the rationale before they show these results. Should make it clearer that the GPR56 signal results after gating in Figure 1E. In addition, the authors note the potential using of GPR56 for differentiating between MG and infiltrating macrophages, but also show that GPR56 is one of the few genes that is downregulated NAWM.

3) In the gating strategy they gate microglia as CD14⁻ and macrophages as CD14⁺ but previous report (PMID: 24316888) shows high expression of CD14 in microglia including additional two manuscripts the authors indicate in their own Supp. Fig.2.

4) Figure 1 may be more appropriate as a supplementary figure

5) Fig 2G. It is interesting that GM microglia from MS subjects upregulate neurodegenerative SPP1 and GPNMB genes which have been shown to upregulated in neurodegenerative microglia (PMID: 28930663), compared to control GM. The authors should discuss it.

Minor comments:

1. Fig 2D: incorrect x-axis label

2. Fig1: The authors should provide IHC image demonstrating Gpr56 expression by microglia (if an appropriate antibody exists).

Reviewer #3 (Remarks to the Author):

Poel et al take advantage of their access to rapid autopsy material to analyze the molecular properties of microglia in white and grey matter of MS cases and control. The emphasis of the MS cases is on normal appearing white matter and non-lesional grey matter. The overall conclusions relate to: i)

regional differences (grey vs white matter) in microglia profiles in “normal” brain and II) similarities and differences between grey and white matter profiles of the MS cases compared to controls and relating these to disease features. The data presented is derived from RNA sequencing of microglia isolated by initial collagenase digestion/Percoll gradient centrifugation followed by immune-magnetic bead separation using anti-CD11b antibody. Patients and controls were in mid 60s-70s with little information (clinical/MRI/pathologic) regarding recent disease activity.

Overall issue regarding what are the novel data to be gained from this study.

i)Regional differences - by use of autopsy tissue, the authors have the capacity to demonstrate differences in grey vs white matter; this was not done in studies using surgically resected tissues (eg Gosselin et al). The latter study used a similar Percoll isolation technique and then isolated cells by FACS that were CD11b+CD45LowCD64+CX3CR1High so as to exclude macrophages that were present “in variable amounts”. The bulk cell sequencing used in the current study did not exclude CD45 high cells. Authors do provide data that microglia linked genes are highly expressed.

Do the authors have any sequencing data on total tissue samples to match the Gosselin et al study to address the contribution of pre-morbid and post-mortem variables. Post-mortem delay is said not to be a significant contributor.

Of interest would be to know if there are differences in microglia properties in different grey matter or white matter regions in controls – ie use of autopsy material (multiple samples from same donor) would allow us to learn about basic microglia heterogeneity.

ii)MS profiles –the focus of the current study is on normal appearing tissue. The use of the autopsy cases could have provided the opportunity to assess the gradation of microglia changes during lesion evolution in MS, especially inclusion early or pre-lesional changes in NAWM and peri-lesional changes (previously characterized immuno-histochemically by these authors).

The authors have the immuno-histochemical techniques in hand to document the status of the microglia in the samples used for sequencing. Were the regions of brain from the MS cases immuno-histochemically comparable to “control” brain”? Do the authors conclude that all the sequencing changes found in MS are consequent to disease related events?

Immuno-histochemistry studies would also address extent of heterogeneity of microglia within the same regions as described in studies of actual MS lesions.

iii)Bio-informatics data – multiple potentially disease relevant pathways (eg lipid metabolism, interferon signaling pathways) are implicated by the bio-informatics analysis but more extensive studies would be needed to confirm these. Current data show limited linkage between gene expression and protein expression.

iv) Donor age - there is some concern about how the donors used in the study (advanced age) reflect initial events in MS – what is consequence of advanced disease and age?

With regard to specific data presented:

-NAWM was defined by MRI – suggest that authors document the status of the sections used for gene expression by immuno-histochemistry as they have done before.

Fig 1 demonstrates that the isolation procedure is enriched in microglia – not quite the same as specifically excluding macrophages such as by CD45 low sorting. What proportion of selected cells were microglia vs macrophages based on flow cytometry analysis? This reviewer appreciates the authors' use of meningeal macrophages as a comparator.

Fig 2 shows more differentially expressed genes between control GM and WM than for GM vs WM in

MS – what is the presumed basis for this; does it represent activation state of these cells? Fig 2D – is this mislabeled? – should it be control not MS?

Bioinformatics – Fig 3 transcriptional networks – a whole series of networks not previously documented to distinguish microglia in GM vs WM in controls and MS are presented. Attempts are made to confirm some of these by protein expression or signaling activity but little confirmation could be obtained (Figs 4 and 5). Would any of these pathways be more apparent at sites of greater pathology and allow for gene/protein correlations?

Mention is made of “high inter-donor variation in gene expression patterns for WM control microglia”. What is variation for Controls and MS if use multiple sites from the same donor tissue?

Some concern whether changes in gene expression patterns (eg complement, inflammatory response related) could reflect age acquired changes.

Point-by-point reply for manuscript “Transcriptional profiling of human microglia reveals grey–white matter heterogeneity and multiple sclerosis-associated changes”

Below we give a point-by-point reply (**R**) to the reviewers’ comments (**C**), followed by any revisions that were made in the manuscript. We thank the reviewers for the time invested in evaluating our manuscript, and appreciate their valuable insights and remarks.

Reviewer #2

Overall an interesting manuscript that is logically and well-written. The findings will serve an important resource for human microglia transcriptome in early stages of MS. However, there are several comments to be addressed before acceptance.

Major Comments:

C1. There is no gene expression profiling of microglia from MS lesions. Thus, the manuscript lacks a comparison between the microglia from normal appearing GM/WM and microglia within the lesions.

R1. For our study, we focused on normal-appearing MS tissue to identify initial events possible preceding MS lesion formation. A complete analysis of the microglial transcriptome from different lesion subtypes lies beyond the scope of this study. However, we agree with the reviewer that information on gene expression in microglia from MS lesions is of importance to relate the here reported microglial changes in NAWM/NAGM to MS lesion pathology. Therefore, we performed additional experiments to assess gene expression levels of lipid metabolism genes, which are upregulated in MS NAWM microglia, in MS lesion tissue as well as microglia nuclear fractions derived from those lesions. IRF8⁺ nuclei were isolated from MS lesions and NAWM by FACS sorting, to investigate gene expression in nuclear fractions enriched for microglia. We show that *CH13L1* and *LPL* expression was increased in and around mixed active/inactive MS lesions in both whole tissue as well as in the IRF8⁺ nuclear fraction. These results clearly confirmed that transcriptional changes in microglia from NAWM of MS relate to transcriptional changes in MS lesion pathology (see the novel Figure 5 and Supplemental Figure 8).

C2. The authors should provide additional information on how normal-appearing GM and WM were defined (all they say is by using MRI guidance). How far away were selected areas (corpus callosum and occipital cortex) from pre-existing MS lesions?

And why did authors choose occipital cortex in particular? Temporal lobes are more commonly affected in MS, and therefore may have been a better site to look for “pre-MS” microglial transcriptional changes. When did these patients take MRI? Which sequences did the authors use? According to line 270, “due to MRI, had a greater post-mortem delay”, it seems that the patients took MRI after they passed away. Lacking the supply of blood and oxygen, will the brain change drastically in MRI? Are the entire corpus callosum or occipital gyrus normal appearing in the MRI? If not, how could the authors get the non-lesion part of these regions? Please clarify it.

R2. At autopsy, normal-appearing tissue was taken out under guidance of post-mortem MRI. Brains of MS donors were cut into 10 mm-thick coronal slices and MRI scanned to determine absence of lesions in corpus callosum and occipital cortex in each slice. Tissue blocks designated as normal-appearing for microglia isolation were collected from scanned brain slices. In addition, a small part of each normal-appearing tissue block was immunohistochemically stained for HLA-DR, CD68, and PLP to determine microglia morphology and myelin integrity (see novel Supplemental Figure 2 for representative immunohistochemically stained images of MS and control white matter tissue).

Occipital cortex was chosen for pragmatic reasons, as availability of large tissue blocks from other cortical regions is very limited. Using occipital cortex therefore ensures the regional consistency needed for microglial profiling.

MRI sequences used were 3D-FLAIR, DIR, T1, and T2.

The time between death and autopsy of donors from the NBB is short (on average 6 hours), thereby minimizing the effect of post-mortem variables on the tissue. Post-mortem MRI analyses of coronal slices takes one hour and has proven to be very useful for histological validation of MS pathology (PMID: 25761376, 15043708 and 29496152).

Corpus callosum and occipital cortex tissue blocks of 4-6 gram were excised from a scanned 10 mm-thick brain slice, so not the entire corpus callosum or occipital cortex was taken out.

We added a short description for post-mortem MRI-guided dissection of 10 mm coronal MS brain slices (see Materials and Methods, page 5).

C3. Why the authors focus on Gpr56? The authors should provide the rationale before they show these results. Should make it clearer that the GPR56 signal results after gating in Figure 1E.

R3. We noticed the abundance of G protein-coupled receptors among microglia signature genes and found that GPR56 is one of the most highly expressed genes in both GM and WM human microglia (Figure 1D). In our previous research, we showed that monocyte-derived macrophages do not express GPR56 (PMID: 28950945), and indeed in the current manuscript, we show that GPR56 can be used to distinguish human microglia from choroid plexus macrophages by flow cytometry (see Figure 1E). We therefore consider GPR56 as an excellent marker to distinguish microglia also from infiltrating monocyte-derived macrophages. In line herewith, a recent mouse study showed that GPR56 indeed distinguishes microglia from macrophages that repopulate the brain under pathological conditions (PMID: 29861285). Our study describes GPR56 as a signature gene of human microglia. To better explain the rationale for focusing on GPR56, we have rephrased the text at page 12-13.

C4. In addition, the authors note the potential using of GPR56 for differentiating between MG and infiltrating macrophages, but also show that GPR56 is one of the few genes that is downregulated NAWM.

R4. Expression of GPR56 is indeed reduced in NAWM as compared to control WM microglia, but still clearly present at both gene and protein level (see Figure 4E) and therefore useful to distinguish microglia from macrophages also in NAWM using flow cytometry.

C5. In the gating strategy they gate microglia as CD14⁻ and macrophages as CD14⁺ but previous report (PMID: 24316888) shows high expression of CD14 in microglia including additional two manuscripts the authors indicate in their own Supp. Fig.2.

R5. We have adjusted the text on the gating strategy for Supplemental Figure 4: we have gated microglia as CD14^{dim} and choroid plexus macrophages as CD14^{high} cells by flow cytometry, as choroid plexus macrophages more highly express CD14 as compared to microglia shortly after isolation (PMID: 24014207). CD14 expression is only increased on primary microglia during culturing, therefore we did not detect high expression of CD14 on microglia acutely profiled after isolation (PMID: 24014207). In the paper referenced by the reviewer (PMID: 24316888), human microglia were isolated by prolonged adherence in culture, possibly explaining the high CD14 expression they detect in brain-derived myeloid cells. The two papers that we refer to in Supplemental Figure 3 both show higher expression of CD14 in microglia when comparing them to whole cortex, but they do not differentiate in gene expression between human microglia as compared to human macrophages.

C6. Figure 1 may be more appropriate as a supplementary figure.

R6. In Figure 1, we present a common human microglia signature for both GM and WM regions and compared expression of these signature genes with macrophages. Furthermore, we focus on *ADGRG1*/*GPR56* as an important generic microglia marker to distinguish microglia from infiltrating macrophages. We consider these as important new findings and essential part of the main narrative of the manuscript, we therefore kept Figure 1 as regular displayed item.

C7. Fig 2G. It is interesting that GM microglia from MS subjects upregulate neurodegenerative *SPP1* and *GPNMB* genes which have been shown to upregulated in neurodegenerative microglia (PMID: 28930663), compared to control GM. The authors should discuss it.

R7. We thank the reviewer for this interesting insight, we now discuss the upregulation of genes in NAGM microglia in relation to the neurodegenerative microglia profile (*SPP1*, *GPNMB*, and *CXCR4*) in more detail on page 21.

Minor comments:

C8. Fig 2D: incorrect x-axis label

R8. We have adjusted the x-axis label of Figure 2D

C9. Fig1: The authors should provide IHC image demonstrating Gpr56 expression by microglia (if an appropriate antibody exists).

R9. We have extensively tried to show GPR56 expression by microglia with IHC by testing four different antibodies, but unfortunately we did not succeed in staining GPR56 on microglia in human tissue.

Poel et al take advantage of their access to rapid autopsy material to analyze the molecular properties of microglia in white and grey matter of MS cases and control. The emphasis of the MS cases is on normal appearing white matter and non-lesional grey matter. The overall conclusions relate to: i) regional differences (grey vs white matter) in microglia profiles in “normal” brain and ii) similarities and differences between grey and white matter profiles of the MS cases compared to controls and relating these to disease features. The data presented is derived from RNA sequencing of microglia isolated by initial collagenase digestion/Percoll gradient centrifugation followed by immune-magnetic bead separation using anti-CD11b antibody.

C10. Patients and controls were in mid 60s-70s with little information (clinical/MRI/pathologic) regarding recent disease activity.

R10. The MS donors in our study had a disease duration of 29 ± 3.5 years and an active lesion load of on average 0.37, meaning that 37% of all lesions were active at the moment of autopsy. Furthermore, three donors were diagnosed with primary-progressive MS and 7 donors had secondary-progressive MS. We have expanded the clinical information to incorporate additional disease activity measures in Table 1.

C11. Overall issue regarding what are the novel data to be gained from this study.

R11. This study is the first to report transcriptional region heterogeneity for human microglia, where GM–WM differences relate to immune regulation. Moreover, we are the first to describe changes in human microglia gene expression in relation to brain disease. We show that transcriptional differences for microglia in normal-appearing MS tissue are region-specific and confirm that the transcriptional changes in NAWM microglia are first signs of MS lesion pathology.

C12. Regional differences – by use of autopsy tissue, the authors have the capacity to demonstrate differences in grey vs white matter; this was not done in studies using surgically resected tissues (eg Gosselin et al). The latter study used a similar Percoll isolation technique and then isolated cells by FACS that were CD11b+CD45LowCD64+CX3CR1High so as to exclude macrophages that were present “in variable amounts”. The bulk cell sequencing used in the current study did not exclude CD45 high cells. Authors do provide data that microglia linked genes are highly expressed.

R12. During microglia isolation, we do not exclude CD45high cells using flow cytometry, since this will exclude activated microglia in normal-appearing MS tissue. Our group previously showed that CD45 expression is higher in MS NAWM microglia as compared to control WM microglia (PMID: 24014207), therefore we did not make a selection for CD45low cells in our study. Furthermore, in contrast to the CD45–CD11b FACS plots of microglia isolated by Gosselin and colleagues, we do not observe a distinct population of CD45+ populations in our CD11b+ bead-captured population for both GM and WM microglia from

post-mortem control and MS tissue, which makes gating for microglia based on CD45 intensity completely arbitrary. See Rebuttal Figure 1 for CD45–CD11b dot plots.

Rebuttal Figure 1: Microglia isolated for RNA-sequencing show one homogenous CD45-CD11b population by flow cytometry. Representative dot plots for microglia populations isolated from GM and WM tissue of control and MS donors used for RNA-sequencing. Dot plots show a homogenous CD45–CD11b population after gating for viable cells. CD45–CD11b protein expression is higher in WM microglia compared to GM microglia.

C13. Do the authors have any sequencing data on total tissue samples to match the Gosselin et al study to address the contribution of pre-morbid and post-mortem variables. Post-mortem delay is said not to be a significant contributor.

R13. We do not have RNA-sequencing data from biopsy tissue to check the effect of post-mortem delay (PMD) on gene expression. However, recently Galatro and colleagues compared resected and post-mortem samples and found no effect of PMD on RNA expression of astrocyte, oligodendrocyte, neuronal, and microglial genes. In earlier studies, we described that PMD in the range of the NBB (4–10 h) does not affect RIN values (PMID: 20010301) nor CD45–CD11b protein expression based on flow-cytometric analyses (PMID: 28212663).

C14. Of interest would be to know if there are differences in microglia properties in different grey matter or white matter regions in controls – ie use of autopsy material (multiple samples from same donor) would allow us to learn about basic microglia heterogeneity.

R14. We fully agree with the reviewer that comparisons of different GM or WM regions is of interest. However, for reasons of capacity we now confined to two regions in MS and controls, but it would certainly be of interest to include more regions for future studies to study human microglia heterogeneity.

C15. MS profiles –the focus of the current study is on normal appearing tissue. The use of the autopsy cases could have provided the opportunity to assess the gradation of microglia changes during lesion evolution in MS, especially inclusion early or pre-lesional changes in NAWM and peri-lesional changes (previously characterized immuno-histochemically by these authors).

R15. We agree with the reviewer that it would be highly interesting to assess gene expression in different gradations of microglia activation in relation to MS pathology. However, in fresh tissue dissected at autopsy, it is not possible to define lesion activity using IHC and to isolate microglia specifically from e.g. the rim and peri-rim of MS lesions. For this reason, we recently successfully developed a method to isolate and sort IRF8⁺ nuclei from frozen MS lesion tissue from which we can analyze nuclear RNA. This technique will make future studies possible to specifically characterize IRF8⁺ microglia enriched nuclear RNA from well-characterized MS lesion tissue to determine gene expression profile of different stages of activation in relation to MS pathology. We added novel data obtained using this approach to the manuscript (see novel Supplemental Figure 8 and Figure 5). We would like to refer reviewer 3 to **C1/R1** above for additional information.

C16. The authors have the immuno-histochemical techniques in hand to document the status of the microglia in the samples used for sequencing. Were the regions of brain from the MS cases immuno-histochemically comparable to “control” brain”?

R16. We agree with the reviewer and have immunohistochemically stained sections from tissue used for microglia isolation, for HLA-DR, CD68, and myelin protein PLP to study microglia morphology, activation, and the possible presence of demyelination. There was variation in intensity and number of HLA-DR- and CD68-stained cells between donors, but the cells showed comparable ramified morphology in all control and MS tissue blocks used for microglia isolation. We noticed no signs of demyelination in any of the blocks. Representative images of an MS and control white matter tissue block were added to the manuscript (see novel Supplemental Figure 2).

C17. Do the authors conclude that all the sequencing changes found in MS are consequent to disease related events? Immuno-histochemistry studies would also address extent of heterogeneity of microglia within the same regions as described in studies of actual MS lesions.

R17. We indeed conclude that changes found in MS normal-appearing microglia are consequent to disease related events:

First, we previously published that expression of normal-appearing MS related genes *CHI3L1*, *CXCR4*, and *GPNMB*, is also significantly increased in the rim and peri-rim regions of chronic active MS lesions as compared to control WM tissue (PMID: 29312322).

Second, we now performed additional experiments to analyze gene expression of *CHI3L1*, *LPL*, *EEPD1*, and *ADGRG1* in IRF8⁺ nuclei from MS lesions and confirm that early changes we found in NAWM microglia are related to MS lesion pathology, as these genes are also expressed or show even higher expression in microglial enriched nuclear fractions from mixed active/inactive MS lesions (also see **R15** and novel Figure 5).

Third, changes we observe in NAWM microglia might be a consequence of Wallerian degeneration, and may suggest an indirect response to an MS lesion situated close by that causes axonal damage (see Discussion, page 21).

Finally, microglia transcriptional changes in NAWM/NAGM are not related to variables like age or post-mortem delay.

We agree that studying microglial heterogeneity within one region is highly interesting, but using IHC relies heavily on the availability of reliable antibodies. It would also be possible using a single nucleus sequencing approach combined with nuclear sorting, but this is not part of the scope of this manuscript.

C18. Bio-informatics data – multiple potentially disease relevant pathways (eg lipid metabolism, interferon signaling pathways) are implicated by the bio-informatics analysis but more extensive studies would be needed to confirm these. Current data show limited linkage between gene expression and protein expression.

R18. In a previous study, we showed expression of proteins involved in lipid processing, CHIT1 and GPNMB, around chronic active MS lesions by immunohistochemistry (PMID: 29312322), confirming lipid metabolism expression at protein level in areas seemingly devoid of MS pathology.

For the current manuscript, we show protein expression of both STAT2 (interferon pathway) and NF- κ B subunits in microglia of control donors, by immunohistochemistry (see Supplemental Figure 5 and 6). In addition, we have extensively tried to stain EEPD1 by IHC in brain tissue, unfortunately the antibody did not work.

C19. Donor age - there is some concern about how the donors used in the study (advanced age) reflect initial events in MS – what is consequence of advanced disease and age?

R19. Our group previously showed that the NBB MS autopsy cohort collected from 1990–2015 has a disease duration of 29 ± 13 years, of which 57% of all lesions (7,562 blocks in total) were mixed active/inactive based on evaluation of HLA/PLP-double stainings (PMID: 29441412). Thus, myeloid inflammatory activity is still high in MS patients with a long disease duration. Seven MS donors for RNA-sequencing analysis also had a chronic progressive disease course and on average, all MS donors had a disease duration of 29 ± 3.5 years. They had an active lesion load of on average 0.37, meaning that 37% of all lesions were active at the moment of autopsy. This is now added to the text at page 12 and in Table 1 and Supplemental Table 1. The MS and control groups used for RNA-sequencing were not significantly different in age nor did we detect a correlation between age and expression of DE genes.

With regard to specific data presented:

C20. NAWM was defined by MRI – suggest that authors document the status of the sections used for gene expression by immune-histochemistry as they have done before.

R20. NAWM tissue was dissected on MRI guidance during autopsy (also see **R2**). We stained tissue of control and MS donors used for microglia isolation with microglia markers HLA-DR,

CD68, and myelin protein PLP as described above (see **R16**). Both control and MS donors showed clear ramified microglia morphology and showed no difference in microglia activation. PLP staining showed no signs of demyelination in normal-appearing MS tissue, confirming the absence of lesions assessed by MRI (see novel Supplemental Figure 2).

C21. Fig 1 demonstrates that the isolation procedure is enriched in microglia – not quite the same as specifically excluding macrophages such as by CD45 low sorting. What proportion of selected cells were microglia vs macrophages based on flow cytometry analysis? This reviewer appreciates the authors' use of meningeal macrophages as a comparator.

R21. We did not observe two separate populations based on CD45 expression, therefore we cannot provide the percentage of microglia and macrophages in our samples. As described previously, we distinguished autologous choroid plexus macrophages and microglia using cell tracker labeling of macrophages. Macrophages had higher CD45 expression as compared to WM microglia (PMID: 28212663), and GM microglia had lower CD45 expression as compared to WM microglia. The average expression of CD45 in the CD11b-bead captured population in this study is lower as compared to the average CD45 expression by macrophages previously described by our group (PMID: 28212663). Also see **R12** and Rebuttal Figure 1.

C22. Fig 2 shows more differentially expressed genes between control GM and WM than for GM vs WM in MS – what is the presumed basis for this; does it represent activation state of these cells?

R22. As compared to control GM microglia, control WM microglia show significant higher expression of several genes related to the NF- κ B pathway, *NFKBIZ* and *NFKBIA*, but microglia isolated from MS WM as compared to MS GM do not show a significantly higher expression of these genes anymore. This indicates that microglia in MS may start losing their region-specific profile.

Furthermore, no difference in the homeostatic transcriptional profile between GM and WM microglia in both control and MS was observed, showing no difference between GM-WM regions in the overall activation status of microglia.

We now mentioned these observations in the Discussion (page 19).

C23. Fig 2D – is this mislabeled? – should it be control not MS?

R23. We adjusted the x-axis label for Figure 2D.

C24. Bioinformatics – Fig 3 transcriptional networks – a whole series of networks not previously documented to distinguish microglia in GM vs WM in controls and MS are presented. Attempts are made to confirm some of these by protein expression or signaling activity but little confirmation could be obtained (Figs 4 and 5).

Would any of these pathways be more apparent at sites of greater pathology and allow for gene/protein correlations?

R24. In control tissue, we hardly observed nuclear presence of NF-κB proteins in WM and GM microglia, which is a sign of NF-κB pathway activation. However, when microglia are triggered by an immune stimulus we would expect a difference between WM and GM microglia in the activation of NF-κB pathway, because expression of NF-κB inhibitor genes is higher in WM compared to GM microglia. Therefore, nuclear translocation of NF-κB proteins will be different in WM microglia, and they will be kept in a more quiescent state upon triggering by an immune stimulus. Region-dependent difference in NF-κB pathway activation in microglia will play an important role in how microglia respond to immune stimuli and might contribute to differences in GM–WM MS lesion pathology, which is just one example of a possible implication of regional microglia differences. Investigating the associated pathways specifically in MS lesion microglia will require an extensive characterization of the microglial transcriptome and proteome derived from lesion tissue.

C25. Mention is made of “high inter-donor variation in gene expression patterns for WM control microglia”. What is variation for Controls and MS if use multiple sites from the same donor tissue?

R25. We mention that gene expression in WM microglia varied between control donors (module darkseagreen, Figure 3), to show that a WM signature for only control donors does not exist. To be clear, for reasons of capacity we isolated microglia from only 2 regions (WM/GM) per donor, so high variation between donors in microglial gene expression is to be expected.

C26. Some concern whether changes in gene expression patterns (eg complement, inflammatory response related) could reflect age acquired changes.

R26. The increased expression of genes related to the complement pathway and inflammation are observed in microglia from control GM tissue. Age is not significantly different between MS and control donors, therefore expression of these genes is unlikely to be affected by age. Finally, age-related microglia genes like *CX3CR1*, *CTSD*, and *P2RY12*, which show reduced expression in donors with an average age of 94 (PMID: 29416036) do not change between control and MS donors in our dataset (Figure 4).

PMID list

PMID: 15043708	Bö, Neuropathol ApplNeurobiol 2004
PMID: 20010301	Durrenberger, JNeuropatholExpNeurol 2010
PMID: 24014207	Melief, Glia 2013
PMID: 24316888	Butovsky, NatNeurosci 2014
PMID: 25761376	Jonkman, JNeurol 2015
PMID: 29312322	Hendrickx, FrontImmunol 2017

PMID: 28930663	Krasemann, Immunity 2017
PMID: 28950945	Lin, AdvImmunol, 2017
PMID: 28212663	Mizee, ANC 2017
PMID: 29861285	Bennet, Neuron 2018
PMID: 29496152	Jonkman and Geurts, HandClinNeurol 2018
PMID: 29441412	Luchetti, ActaNeuropathol 2018
PMID: 29416036	Olah, NatCommun 2018

REVIEWERS' COMMENTS:

Reviewer #2 (Remarks to the Author):

The authors addressed all the Reviewer's comments. I recommend acceptance of the manuscript.

Reviewer #3 (Remarks to the Author):

The revised manuscript by Poel et al addresses, as best as possible, the issues raised in the review. The issues raised included both specific items regarding patient selection, technical items, requests for additional information regarding MS pathology. The manuscript and letter of response provide clear information regarding patient selection (rather chronic disease but significant abundance of active lesions are documented). The authors also supply the rationale for selection of specific tissue sites based on topography and MRI and lack of concern about post-mortem delay. This data is sufficient to allow others to compare their data with this report.

The authors present data on GPR56 as a marker of microglia vs macrophages. The NAWM isolated myeloid cells do differ from meningeal macrophages. However, as the authors indicate, GPR56 is down-regulated in NAWM vs control white matter and in vitro expression is completely lost. Thus one is still concerned about using GPR56 as a microglia marker under conditions where both macrophages and microglia are present and where they may be highly activated. It is disappointing that the authors could not use GPR 56 immunohistochemistry and combine this with other microglia markers. For the current study, one would accept that most of the myeloid cells in NAWM would be microglia.

As regards the request for more information regarding MS pathology, one agrees with the author that one article cannot cover all the issues. The revised manuscript documents that there is microglia activation in NAWM and there is no active demyelination (no myelin in the myeloid cells). They do provide some additional data regarding similar up-regulation of lipid metabolism genes in NAWM and actual lesions.

Overall the data make a significant contribution to defining microglia/macrophage pathology in MS taking advantage of material that is difficult to access. The limitations of the study are realistically presented.

Point-by-point reply for manuscript “Transcriptional profiling of human microglia reveals grey–white matter heterogeneity and multiple sclerosis-associated changes”

Below we reply (**R**) to the reviewers’ comments (**C**), followed by any revisions that were made in the revised manuscript.

REVIEWERS’ COMMENTS:

Reviewer #2 (Remarks to the Author):

C. The authors addressed all the Reviewer's comments. I recommend acceptance of the manuscript.

R. We thank the reviewer for evaluating our revised manuscript and his/her advice to accept our revised manuscript for publication in Nature Communications.

Reviewer #3 (Remarks to the Author):

C1. The revised manuscript by Poel et al addresses, as best as possible, the issues raised in the review. The issues raised included both specific items regarding patient selection, technical items, requests for additional information regarding MS pathology. The manuscript and letter of response provide clear information regarding patient selection (rather chronic disease but significant abundance of active lesions are documented). The authors also supply the rationale for selection of specific tissue sites based on topography and MRI and lack of concern about post-mortem delay This data is sufficient to allow others to compare their data with this report.

R1. We thank the reviewer for the compliments and we are happy to read that we addressed the issues regarding patient selection adequately.

C2. The authors present data on GPR56 as a marker of microglia vs macrophages. The NAWM isolated myeloid cells do differ from meningeal macrophages. However, as the authors indicate, GPR56 is down-regulated in NAWM vs control white matter and in vitro expression is completely lost. Thus one is still concerned about using GPR56 as a microglia marker under conditions where both macrophages and microglia are present and where they may be highly activated. It is disappointing that the authors could not use GPR56 immunohistochemistry and combine this with other microglia markers. For the current study, one would accept that most of the myeloid cells in NAWM would be microglia.

R2. The expression of GPR56 is indeed downregulated in NAWM microglia of MS donors, but the expression is still higher (mean of 1,733, see Figure 4E) as compared to choroid plexus macrophages (mean of 514, see Figure 1E), therefore we suggest that GPR56 still distinguishes microglia from macrophages in NAWM tissue. We agree with the reviewer that more research is needed to understand the expression of *ADGRG1*/GPR56 under pathological conditions where macrophages and microglia are both present. However, we show that *ADGRG1* expression is still present in IRF8⁺ nuclei isolated from chronic active lesions (see Figure 5B), suggesting that GPR56 might be a useful marker to distinguish microglia from infiltrating macrophages in chronic active MS lesions. We have now added two extra sentences on GPR56 expression on microglia in active MS lesions in the Discussion on page 13.

We have extensively tried to stain microglia for GRP56 with four different antibodies in brain tissue, but unfortunately were not able to detect GPR56 expression by microglia. However, we succeeded to

detect GPR56 expression on acute isolated microglia with both Western blot and flow cytometry and show that the GPR56 antibody can be used to distinguish acute isolated microglia from macrophages.

C3. As regards the request for more information regarding MS pathology, one agrees with the author that one article cannot cover all the issues. The revised manuscript documents that there is microglia activation in NAWM and there is no active demyelination (no myelin in the myeloid cells). They do provide some additional data regarding similar up-regulation of lipid metabolism genes in NAWM and actual lesions.

R3. We highly appreciate the reviewer comments on the data we have added showing microglia morphology in normal-appearing tissue and gene expression information regarding MS pathology.

C4. Overall the data make a significant contribution to defining microglia/macrophage pathology in MS taking advantage of material that is difficult to access. The limitations of the study are realistically presented.

R4. We thank the reviewer for the favorable evaluation of our manuscript .